# Think-Then-Generate: Reasoning-Aware Text-to-Image Diffusion with LLM Encoders

Siqi Kou [* 1 2]   Jiachun Jin [* 1 2]   Zetong Zhou [* 1]   Ye Ma [2]   Yugang Wang [1]   Quan Chen [2]   Peng Jiang [2]
Xiao Yang [3]   Jun Zhu [3]   Kai Yu [1]   Zhijie Deng [1]

## Abstract

Recent progress in text-to-image (T2I) diffusion models (DMs) has enabled high-quality visual synthesis from diverse textual prompts. Yet, most existing T2I DMs, even those equipped with large language model (LLM) based text encoders, remain text–pixel mappers—they employ LLMs merely as text encoders, without leveraging their inherent reasoning capabilities to infer what should be visually depicted given the textual prompt. To move beyond such literal generation, we propose the **think-then-generate (T2G)** paradigm, where the LLM text encoder is encouraged to reason about and rewrite raw user prompts; the states of the rewritten prompts then serve as diffusion conditioning. To achieve this, we first activate the T2G pattern of the LLM encoder with a lightweight supervised fine-tuning process. Subsequently, the LLM encoder and diffusion backbone are co-optimized to ensure faithful reasoning about the context and accurate rendering of the semantics via **Dual-GRPO**. In particular, the text encoder is reinforced using image-grounded rewards to infer and recall world knowledge, while the diffusion backbone is pushed to produce semantically consistent and visually coherent images. Experiments show substantial improvements in semantic alignment and visual realism on reasoning-based image generation and editing benchmarks, achieving 0.79 on WISE score, nearly on par with GPT-4o. Our results pave the way for next-generation unified models with reasoning, expression, and demonstration capacities. Our code is available at https://github.com/zhijie-group/think-then-generate.

*Equal contribution. Work done during Siqi and Jiachun's internships at Kuaishou Technology. [1]Shanghai Jiao Tong University [2]Kuaishou Technology [3]Tsinghua University. Correspondence to: Zhijie Deng <zhijied@sjtu.edu.cn>.

*Proceedings of the 43rd International Conference on Machine Learning*, Seoul, South Korea. PMLR 306, 2026. Copyright 2026 by the author(s).

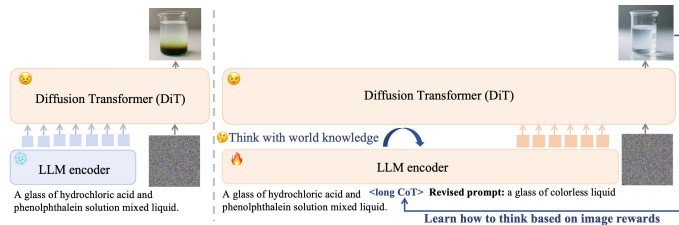

*Figure 1.* An overview of our *T2G* method. Beyond using LLM as a frozen text encoder, we train it to think and refine raw user prompts guided by image-grounded rewards. This enables aligning textual reasoning with visual generation in an end-to-end manner.

## 1. Introduction

Text-to-image (T2I) diffusion models (DMs) have demonstrated remarkable capabilities in generating high-fidelity and diverse images given textual prompts, as exemplified by Stable Diffusion (Rombach et al., 2022; Esser et al., 2024) and the FLUX series (Labs, 2023; Batifol et al., 2025; Labs et al., 2025). To capture richer semantic representations of textual prompts, researchers have progressively adopted stronger text encoders—from early CLIP models (Rombach et al., 2022) to large language models (LLMs) (Chen et al.; 2024) and their variants with visual inputs, i.e., vision language models (VLMs). Representative examples include the open-source Qwen-Image (Wu et al., 2025), which employs Qwen2.5-VL (Bai et al., 2025) as the encoder, and advanced commercial systems such as Seedream 4.0 (Seedream et al., 2025), GPT-4o (Hurst et al., 2024), and Gemini-2.5-Flash-Image (Google, 2025). The LLM encoder offers two key advantages: (1) native understanding of mixed text–image prompts, and (2) extensive world knowledge that enhances instruction following. These capabilities may alleviate the burden of iteratively crafting and refining prompts, as in prior work (Team, 2024; Mo et al., 2024; Xiang et al., 2025), potentially enabling more natural interactions for visual generation.

However, in practice, existing models often fail to fully exploit the potential of LLMs for reasoning. Typically, the models are trained on large-scale descriptive image–caption pairs with the LLM encoder frozen (Batifol et al., 2025; Wu et al., 2025; Cai et al., 2025) (i.e., solely as a feature extractor). Consequently, they can handle only literal and de-

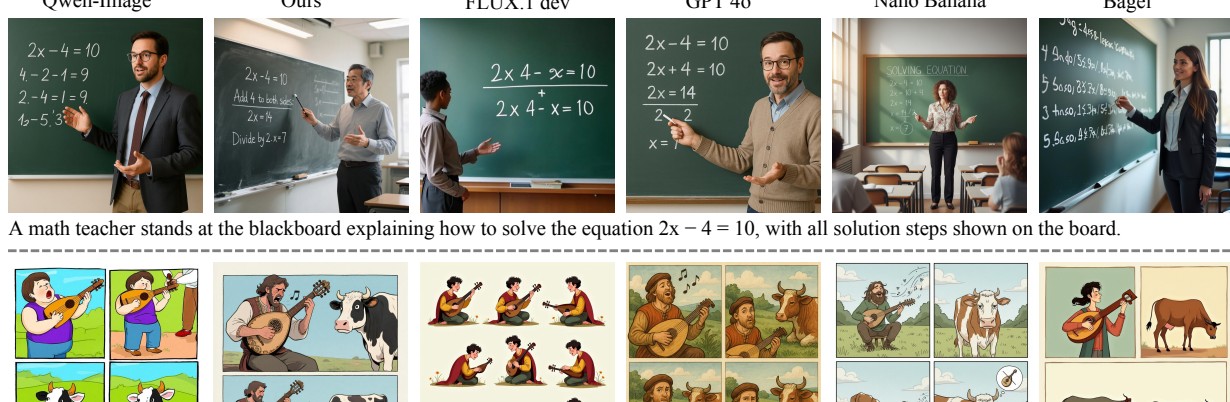

| Qwen-Image | Ours | FLUX.1 dev | GPT 4o | Nano Banana | Bagel |

A math teacher stands at the blackboard explaining how to solve the equation $2x − 4 = 10$, with all solution steps shown on the board.

A multi-panel illustration showing the story of playing the lute to a cow, with steps from passionate perfomance to the cow remaining uninterested.

*Figure 2.* **Comparison of T2I models on conceptual visual generation.** Our Qwen-Image under *T2G* pipeline produces semantically aligned and visually coherent results, correctly interpreting user intent. For example, given the prompt "a multi-panel illustration of playing the lute to a cow", it generates a scene where the cow is uninterested (e.g., just eating grass), rather than literally depicting a lute and a cow, whereas vanilla Qwen-Image behaves like a simple text–pixel mapper and often fails to capture conceptual meanings.

scriptive instructions (*e.g.*, specific object colors or textures) but struggle with conceptual instructions (*e.g.*, illustrating how to solve an equation), as shown in Figure 2. I.e., they function as simple *text–pixel mappers*. Ideally, the LLM encoder should transcend prompt encoding to reason over raw instructions, leverage inherent world knowledge for conceptual tasks, and generate semantically enriched prompts for diffusion generation. However, such a functionality cannot be trivially incentivized by current post-training frameworks for DMs (Liu et al., 2025; Zheng et al., 2025; Yan et al., 2025; Li et al., 2025).

This paper implements the *think-then-generate (T2G)* paradigm for reasoning-aware text-to-image diffusion, where the LLM encoder is able to reason over and rewrite raw user prompts, with the embeddings of rewritten prompts fed to the diffusion transformer (DiT) (Peebles & Xie, 2023; Esser et al., 2024) as generation conditioning, as shown in Figure 1. To realize it, we first construct a supervised fine-tuning (SFT) dataset that enriches raw user prompts with chain-of-thought (CoT) reasoning as well as rewritten prompts. We finetune the LLM on it to acquire the think-then-rewrite pattern. We then co-optimize the LLM encoder and the DiT decoder via a *Dual-GRPO* strategy, where the rewritten prompts act as the bridge between text reasoning and image synthesis and image-based rewards are maximized in an end-to-end manner. Considering the distinct roles of the encoder and the DiT for image generation, we tailored the reward objective to each component, where the former is optimized for *semantic alignment and conceptual understanding* and the latter focuses on *visual realism and aesthetic quality*. This optimization not only activates the world knowledge embedded in LLMs but also adapts DiT to the evolved representation space of the text encoder.

Empirically, on the T2I task, we apply our method to the state-of-the-art open-source model Qwen-Image (Wu et al., 2025), and observe a score of 0.79 on the WISE benchmark (Niu et al., 2025), which surpasses the pre-trained Qwen-Image by 30%, and substantially outperforms other open-source models such as Bagel (Deng et al., 2025) and Janus-Pro-7B (Chen et al., 2025b). Notably, this performance is on par with the commercial GPT-4o (Hurst et al., 2024). Moreover, our method achieves a score of 92.2 on T2I-ReasonBench (Sun et al., 2025), surpassing the strong closed-source model Gemini-2.0 (Google, 2024). For image editing, we apply our method to Qwen-Image-Edit (Wu et al., 2025), achieving a score of 73.4 on UniREdit (Han et al., 2025) and outperforming the closed-source Seedream-4.0 (Seedream et al., 2025). Moreover, in more challenging scenarios—such as generating schematic illustrations of human activities (*e.g.*, teaching math and physics)—our trained model demonstrates superior knowledge grounding, visual plausibility, and aesthetic quality, highlighting its potential as next-generation models with reasoning and demonstration capacities in real-world applications.

## 2. Preliminary: Group Relative Policy Optimization

Group Relative Policy Optimization (GRPO) is a reinforcement learning (RL) algorithm popularized in optimizing large generative models (Shao et al., 2024; Guo et al., 2025). It can be viewed as a variant of policy gradient methods such as PPO (Schulman et al., 2017), with introduced group-wise relative normalization of rewards to eliminate the need for a critic model and improve efficiency significantly.

Formally, given a group of rollout trajectories $\{\mathbf{o}_g\}_{g=1}^G$ sampled from the current policy $\pi_\theta$, unlike actor–critic methods, GRPO groups samples with similar input prompts or conditions, and then directly calculates relative advantages from normalized group rewards rather than learning them through a value model:

$$\hat{A}_g = \frac{R_g - \text{mean}\left(\{R_g\}_{g=1}^G\right)}{\text{std}\left(\{R_g\}_{g=1}^G\right)}, \tag{1}$$

where $R_g$ is the scalar reward for the trajectory $\mathbf{o}_g$. This design eliminates the need for a value model, making GRPO more efficient and easier to scale for large models such as LLMs (Havrilla et al., 2024) and diffusion/flow matching models (Liu et al., 2025; Xue et al., 2025). The typical GRPO objective is:

$$\mathcal{J}_{GRPO}(\theta) = \mathbb{E}_{\{\mathbf{o}_g\}_{g=1}^G \sim \pi_{\theta_{old}}} \tag{2}$$

$$\left[\frac{1}{G}\sum_{g=1}^G \min\left(r_g(\theta)\hat{A}_g, \text{clip}(r_g(\theta), 1-\epsilon, 1+\epsilon)\hat{A}_g\right)\right]$$

$$- \beta \mathbb{D}_{KL}\left[\pi_\theta \parallel \pi_{\theta_{ref}}\right],$$

where $r_g(\theta) = \frac{\pi_\theta(\mathbf{o}_g)}{\pi_{\theta_{old}}(\mathbf{o}_g)}$, $\pi_{\theta_{ref}}$ is the reference policy, and $\beta$ is the KL divergence regularization parameter.

**GRPO for LLMs.** For LLMs, the policy $\pi_\theta(o_t \mid o_{<t}, q)$ is a language model that generates text tokens $o_t$ conditioned on the previous tokens $o_{<t}$ and the user prompt $q$, this holds a tractable likelihood and is easy to compute. In typical GRPO applications, training relies on outcome-based rewards, where a reward model evaluates only the final generated text. All intermediate tokens in the rollout trajectory are assigned the same reward, which implicitly assumes that each token contributes equally to the final outcome.

**Flow-GRPO for Flow Matching Models.** Although GRPO has been successfully applied to LLMs, applying it to flow matching models is nontrivial due to the lack of stochasticity in their ODE-based sampling dynamics, which limits trajectory diversity for advantage estimation and policy exploration. Flow-GRPO (Liu et al., 2025) addresses this by transforming the deterministic Flow-ODE into an equivalent stochastic differential equation (SDE) and discretizing it via the Euler-Maruyama scheme. This yields the following transition kernel:

$$\pi_\theta(\mathbf{x}_{t-1} \mid \mathbf{x}_t) = \mathcal{N}(\mathbf{x}_{t-1}; \mu_\theta(\mathbf{x}_t), g_t^2 \Delta t \mathbf{I}), \tag{3}$$

here $g_t = a\sqrt{\frac{t}{1-t}}$ controls the level of stochasticity, $\Delta t$ is the discretization step size and $\mu_\theta(\mathbf{x}_t)$ equals to:

$$\mathbf{x}_t + \left[\mathbf{v}_\theta(\mathbf{x}_t, t) + \frac{g_t^2}{2t}(\mathbf{x}_t + (1-t)\mathbf{v}_\theta(\mathbf{x}_t, t))\right]\Delta t, \tag{4}$$

where $\mathbf{v}_\theta(\mathbf{x}_t, t)$ is the target velocity field learned via the flow matching objective. Crucially, the transition kernels are

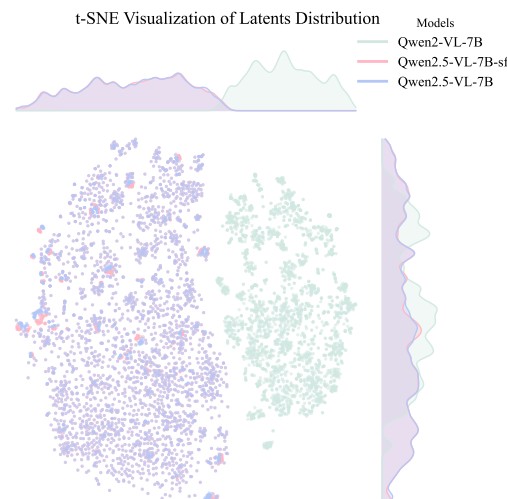

*Figure 3.* **t-SNE visualization of the last layer's hidden state distributions in the LLM encoder before and after SFT.** The distributions remain highly consistent, indicating that our SFT procedure preserves the latent space structure, enabling compatibility with the DiT to render stable and coherent visual outputs.

reduced to tractable Gaussian distributions, which enables the direct application of the GRPO policy update to flow matching models.

## 3. Method

In this section, we first describe how to construct the SFT dataset to train the LLM involved in existing T2I DMs to replicate the *T2G* pattern. We then introduce Dual-GRPO, a reinforcement learning strategy that jointly optimizes both the LLM encoder and the DiT using image-based rewards.

### 3.1. Reasoning-aware Behavior Activation

Existing DMs can be viewed as an LLM-DiT composite model parameterized by $\theta = \{\phi, \lambda\}$, where $p_\phi$ denotes the LLM text encoder and $p_\lambda$ represents the DiT. Given a raw user prompt $q \in Q$ with $Q$ as the prompt set, T2G demands $p_\phi$ to first perform CoT reasoning using its world knowledge to explicitly outline the content to be depicted, followed by summarizing this CoT into a descriptive refined prompt. The last layer's hidden states in $p_\phi$ for the refined prompt are subsequently fed into $p_\lambda$, serving as the conditioning input to generate images. To learn this pattern, we process $Q$ to construct an SFT dataset using Gemini-2.5 (Google, 2025), letting it descriptively infer what should be depicted and then generate a refined prompt. This curated dataset follows the format: raw user prompt $\rightarrow$ [long CoT] $\rightarrow$ refined prompt. The specific instructions for Gemini and an example training data are provided in Appendix A. We then fine-tune $p_\phi$ on this curated dataset.

$p_\phi$ serves as both a rewriter and an encoder in our *T2G* paradigm. While SFT activates its rewriting potential, it

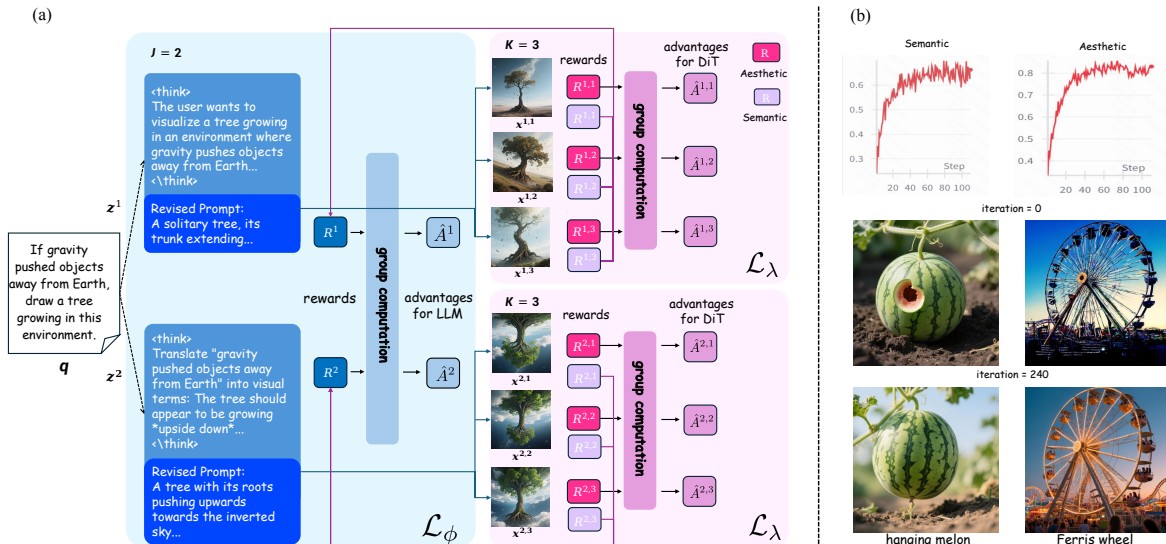

*Figure 4.* Dual-GRPO training trajectories. (a) Tree-structured rollout for a given user prompt $q$: the LLM encoder samples $J$ reasoning traces, each rewritten prompt conditions the DiT to generate $K$ images. Image-grounded rewards are aggregated to compute group-relative advantages for updating both the LLM and the DiT. (b) Evolution of semantic and aesthetic scores during Dual-GRPO training.

remains unclear whether SFT degrades its encoding function: *does SFT alter the latent space in a way that harms the DiT's ability to generate coherent images?* To examine this, we visualize distributions of the last layer's hidden states in Qwen2.5-VL (Bai et al., 2025) before and after SFT using t-SNE (Cai & Ma, 2022). It reveals distributional shifts by showing whether the latents overlap or separate in a low-dimensional space. Interestingly, Figure 3 shows that the latents totally overlap, indicating that SFT preserves the distribution of the embedding space and thereby the DiT can still work.

### 3.2. Aligning Textual Reasoning with Visual Generation via Dual-GRPO

Training $p_\phi$ solely on the SFT dataset without sensing $p_\lambda$ may lead to suboptimal performance. This is because CoT may be flawed, and even if it is correct, the refined prompt expression may not be suitable for $p_\lambda$ to render effectively. This issue is observed in Appendix E. To address this, we propose a Dual-GRPO strategy that jointly optimizes $p_\phi$ and $p_\lambda$ using image-grounded rewards, thereby enabling the alignment of text reasoning with visual generation in an end-to-end framework.

Specifically, let $\mathbf{z} = [z_1, \ldots, z_\ell]$ denote the sequence of $\ell$ generated text tokens (*i.e.,* a combination of CoT reasoning and the refined prompt) from $p_\phi$, and $\mathbf{x} = [x_1, \ldots, x_m]$ represent the reverse-time trajectories of the generated image from $p_\lambda$ with $m$ denoising steps. The rollout trajectory for each generated image can be represented as

$$\mathbf{o} = [z_1, \cdots, z_\ell, x_{\ell+1}, \cdots, x_{\ell+m}].$$

The policy model for DMs is then defined as

$$\pi_\theta(o_t \mid s_t) = \begin{cases} p_\phi(z_t \mid z_{<t}, q) & t \leq \ell, \\ p_\lambda(x_t \mid x_{t-1}, \hat{\mathbf{z}}) & t > \ell, \end{cases} \quad (5)$$

where $\hat{\mathbf{z}}$ denotes the last layer's hidden states in $p_\phi$ for the refined prompt.

To optimize $\pi_\theta$, we maximize the following objective:

$$\max_{\theta=\{\phi,\lambda\}} \mathbb{E}_{q \sim Q, [z_1, \cdots, x_{\ell+m}] \sim \pi_\theta} \quad (6)$$

$$\frac{1}{\ell+m} \left[ \sum_{t=1}^{\ell} R_1(z_t, z_{<t}, q) + \sum_{t=\ell+1}^{\ell+m} R_2(x_t, x_{t-1}, \hat{\mathbf{z}}) \right],$$

where $R_1$ and $R_2$ respectively denote the reward functions for the textual reasoning and denoising trajectories. Unlike the standard RL for LLMs, where a shared outcome-based reward is applied to all intermediate states (Shao et al., 2024), we assign distinct rewards considering different roles of $p_\phi$ and $p_\lambda$ in generating final images. Detailed definitions about the reward functions are provided in Section 3.3.

Given the efficiency gained by omitting the critic model in GRPO (Shao et al., 2024; Liu et al., 2025), we opt for this policy optimization strategy. To compute group-relative advantages, we employ a hierarchical sampling strategy. As illustrated in Figure 4, given a user prompt $q$, we sample $J$ texual sequences $\{\mathbf{z}^j\}_{j=1}^J$ from $p_{\phi_{old}}$. For each $\hat{\mathbf{z}}^j$, we sample $K$ images $\{\mathbf{x}^{j,k}\}_{k=1}^K$ from $p_{\lambda_{old}}$. This procedure

yields the Dual-GRPO objective:

$$\max_{\theta=\{\phi,\lambda\}} \mathbb{E}_{q \sim Q, \{\mathbf{z}^j\}_{j=1}^J \sim p_{\phi_{old}}, \{\mathbf{x}^{j,k}\}_{k=1}^K \sim p_{\lambda_{old}}} \quad (7)$$

$$\frac{1}{\ell+m} \left[ \sum_{t=1}^{\ell} \mathcal{L}_\phi(t, \mathbf{z}^j) + \sum_{t=\ell+1}^{\ell+m} \mathcal{L}_\lambda(t, \mathbf{x}^{j,k}) \right],$$

where $\mathcal{L}_\phi(t, \mathbf{z}^j)$ follows the standard GRPO formulation for LLMs with group size $J$, and $\mathcal{L}_\lambda(t, \mathbf{x}^{j,k})$ is a formulation of flow-GRPO with group size $K$ an extra batch dimension with size $J$.

Specifically, let $\mathbf{z}_t^j$ denote the $t$-th token in $\mathbf{z}^j$ and $\mathbf{z}_{<t}^j$ represent all tokens preceding the $t$-th token. We have:

$$\mathcal{L}_\phi(t, \mathbf{z}^j) = \frac{1}{J} \sum_{j=1}^{J} \left[ \min \left( r_t^j(\phi), \mathrm{clip}(r_t^j(\phi), 1-\epsilon, 1+\epsilon) \right) \hat{A}_t^j \right] \quad (8)$$

$$- \beta \mathbb{D}_{KL} \left[ p_\phi(\mathbf{z}_t^j) \parallel p_{\phi_{ref}}(\mathbf{z}_t^j) \right],$$

where $r_t^j(\phi) = \frac{p_\phi(\mathbf{z}_t^j | \mathbf{z}_{<t}^j, q)}{p_{\phi_{old}}(\mathbf{z}_t^j | \mathbf{z}_{<t}^j, q)}$, and $\hat{A}_t^j$ is calculated by normalizing the group-level rewards as follows:

$$\hat{A}_t^j = \frac{R_1(\mathbf{z}_t^j, \mathbf{z}_{<t}^j, q) - \mathrm{mean}\left( \{R_1(\mathbf{z}_t^j, \mathbf{z}_{<t}^j, q)\}_{j=1}^J \right)}{\mathrm{std}\left( \{R_1(\mathbf{z}_t^j, \mathbf{z}_{<t}^j, q)\}_{j=1}^J \right)}. \quad (9)$$

Correspondingly, $\mathcal{L}_\lambda(t, \mathbf{x}^{j,k})$ is given by:

$$\frac{1}{J} \sum_{j=1}^{J} \frac{1}{K} \sum_{k=1}^{K} \left[ \min \left( r_t^{j,k}(\lambda), \mathrm{clip}(r_t^{j,k}(\lambda), 1-\epsilon, 1+\epsilon) \right) \hat{A}_t^{j,k} \right] \quad (10)$$

$$- \beta \mathbb{D}_{KL} \left[ p_\lambda(\mathbf{x}_t^{j,k} | \mathbf{x}_{t-1}^{j,k}, \hat{\mathbf{z}}^j) \parallel p_{\lambda_{ref}}(\mathbf{x}_t^{j,k} | \mathbf{x}_{t-1}^{j,k}, \hat{\mathbf{z}}^j) \right],$$

where $r_t^{j,k}(\lambda) = \frac{p_\lambda(\mathbf{x}_t^{j,k} | \mathbf{x}_{t-1}^{j,k}, \hat{\mathbf{z}}^j)}{p_{\lambda_{old}}(\mathbf{x}_t^{j,k} | \mathbf{x}_{t-1}^{j,k}, \hat{\mathbf{z}}^j)}$ and $\hat{A}_t^{j,k}$ is the advantage based on $R_2(\mathbf{x}_t^{j,k}, \mathbf{x}_{t-1}^{j,k}, \hat{\mathbf{z}}^j)$.

### 3.3. The Reward Function and Scheduler

In our *T2G* paradigm, $p_\phi$ functions as an interpreter that interprets the intent behind raw user prompts and generates explicit and detailed textual descriptions, while $p_\lambda$ acts as a visual renderer, faithfully mapping textual descriptions into corresponding images. This design necessitates distinct rewards for $p_\phi$ and $p_\lambda$: the former emphasizes semantic alignment and consistency, while the latter focuses on visual realism and aesthetic quality.

Specifically, the reward for $p_\phi$ is the average of $R_{sem}$, which measures semantic consistency and conceptual alignment between the generated images and the raw user prompt. The averaging is done over all $K$ generated images. Formally, it

is defined as:

$$R_1(\mathbf{z}_t^j, \mathbf{z}_{<t}^j, q) = \beta_1(\tau) \frac{1}{K} \sum_{k=1}^{K} R_{sem}(\mathbf{x}^{j,k}, q), \quad (11)$$

where $\tau$ is the current training step and $\beta_1(\tau)$ is a scheduler for reward weighting. The reward for $p_\lambda$ is defined as a weighted sum of the aesthetic score $R_{aes}$ and the physical consistency score $R_{phy}$ of the generated image:

$$R_2(\mathbf{x}_t^{j,k}, \mathbf{x}_{t-1}^{j,k}, \hat{\mathbf{z}}^j) = \beta_2(\tau) \left( \omega_1 R_{aes}(\mathbf{x}^{j,k}) + \omega_2 R_{phy}(\mathbf{x}^{j,k}) \right), \quad (12)$$

where $\omega_1, \omega_2$ are weighting factors and $\beta_2(\tau)$ is also a reward scheduler. This ensures $p_\lambda$ continues to generate visually coherent images given the evolved $\hat{\mathbf{z}}^j$.

## 4. Experiments

In this section, we evaluate the effectiveness of our *T2G* and Dual-GRPO methods across T2I and image editing tasks. Quantitative and qualitative results validate the advantages of our approach.

### 4.1. Implementation Details

We choose to implement *T2G* on the prevalent model Qwen-Image (Wu et al., 2025) and Qwen-Image-edit (Wu et al., 2025) for the T2I and image-editing tasks, respectively. Consequently, the LLM encoder is initialized from Qwen2.5-VL (Bai et al., 2025), and the corresponding DiT backbone is initialized from MM-DiT (Esser et al., 2024). Our post-training pipeline is separated into two stages. For the SFT stage, each model is trained with a learning rate of 5e-6 and a batch size of 32. In the subsequent Dual-GRPO stage, we apply a simple reward scheduler with constant and balanced weights: $\beta_1(\tau) = \beta_2(\tau) = 0.5$, where both the LLM and the DiT are updated jointly at every iteration. In Appendix C, we ablate different designs of the scheduler. During the LLM training, we employ a learning rate of 2e-6 with a batch size of 256, generating 5 responses per input. For the DiT training, we adhere to the default configuration of FlowGRPO-fast(Liu et al., 2025), utilizing a learning rate of 3e-4, a batch size of 32, and a clipping range of 1e-4. The generation process involves sampling 16 images for each prompt over 10 inference steps, with an SDE window of 2. For the reward functions, we use VLM-as-a-judge, following previous work on image rewards (Wang et al., 2025), and adopt Qwen3-30B-A3B (Yang et al., 2025) as the judging model.

### 4.2. Text-to-Image Generation

#### 4.2.1. SUPERVISED FINE-TUNING

For raw user prompt set for T2I, we select R2I-Bench (Chen et al., 2025a), comprising 3k curated prompts that require

*Table 1.* **Comparison of image generation models on WISE.** Numbers in bold indicate the highest score among open-source models. Underlined numbers denote the highest score.

| Model | Type | Cultural | Time | Space | Biology | Physics | Chemistry | Overall ↑ |
|---|---|---|---|---|---|---|---|---|
| FLUX.1-dev | *diffusion* | 0.48 | 0.58 | 0.62 | 0.42 | 0.51 | 0.35 | 0.50 |
| SD-3.5-medium | *diffusion* | 0.43 | 0.50 | 0.52 | 0.41 | 0.53 | 0.33 | 0.45 |
| SD-3.5-large | *diffusion* | 0.44 | 0.50 | 0.58 | 0.44 | 0.52 | 0.31 | 0.46 |
| Bagel w/CoT | *unified* | 0.76 | 0.69 | 0.75 | 0.65 | 0.75 | 0.58 | 0.70 |
| Janus-Pro-7B | *unified* | 0.30 | 0.37 | 0.49 | 0.36 | 0.42 | 0.26 | 0.35 |
| HunyuanImage-3.0 | *unified* | 0.58 | 0.57 | 0.72 | 0.56 | 0.68 | 0.35 | 0.58 |
| T2I-R1 | *unified* | 0.56 | 0.55 | 0.63 | 0.54 | 0.55 | 0.30 | 0.54 |
| Uni-CoT | *unified* | 0.76 | 0.70 | 0.76 | 0.73 | 0.81 | 0.73 | 0.75 |
| GPT 4o | *proprietary* | 0.81 | 0.71 | 0.89 | 0.83 | 0.79 | 0.74 | 0.80 |
| Qwen-Image | *diffusion* | 0.62 | 0.63 | 0.78 | 0.55 | 0.67 | 0.35 | 0.61 |
| Ours (w/o SFT&GRPO) | *diffusion* | 0.68 | 0.58 | 0.77 | 0.62 | 0.76 | 0.41 | 0.65 |
| Ours (w/o GRPO) | *diffusion* | 0.76 | 0.66 | 0.79 | 0.74 | 0.84 | 0.65 | 0.74 |
| Ours | *diffusion* | **0.80** | **0.74** | **0.83** | **0.81** | **0.85** | **0.66** | **0.79** |

integration of world knowledge and reasoning to generate semantically coherent corresponding images. We use this dataset as a template and expand it to 7k prompts by instructing Gemini-2.5 (Google, 2025) for additional instances. Details on the augmentation are provided in Appendix B. We then collect the CoT reasoning and descriptive refined prompts, following Section 3.1, to construct the complete SFT dataset. Qwen2.5-VL is then fine-tuned on this dataset to replicate the *T2G* pattern.

### 4.2.2. QUANTITATIVE RESULTS

**Benchmarks.** We evaluate performance across various reasoning-centric T2I benchmarks. WISE (Niu et al., 2025) evaluates models with 1,000 prompts across cultural common sense, spatio-temporal understanding, and natural science. T2I-ReasonBench (Sun et al., 2025) includes 800 meticulously crafted prompts across four dimensions: idiom interpretation, textual image design, entity reasoning, and scientific reasoning. The first two dimensions, focusing on imagination and information completion, are unique and not included in other benchmarks in the field. It features a two-stage evaluation: an LLM generates question-criterion pairs to check for key reasoning elements, and a multimodal LLM scores the image against these criteria.

**Baselines.** We compare with 10 state-of-the-art T2I models, including 4 diffusion-based T2I models, 4 unified multimodal models, and 2 proprietary models. The diffusion-based T2I models are FLUX.1-dev (Labs, 2023), Stable-Diffusion-3-Medium (Esser et al., 2024), Stable-Diffusion-3.5-Large (Esser et al., 2024), and Qwen-Image (Wu et al., 2025). The unified multimodal models are Bagel (Deng et al., 2025), Uni-CoT (Qin et al., 2025), Emu3 (Wang et al., 2024), Janus-Pro-7B (Chen et al., 2025b), HunyuanImage-3.0 (Cao et al., 2025), and T2I-R1 (Jiang et al., 2025). The proprietary models include Gemini-2.0 (Google, 2024) and GPT-4o (Hurst et al., 2024).

**Results analysis.** From Table 1 and 2, we find that DMs perform poorly on both the WISE and T2I-ReasonBench benchmarks. For example, vanilla Qwen-Image achieves only 0.35 in WISE chemistry domains. This results from a fundamental misalignment between their architecture (treating LLMs only as a text encoder) and the demands of knowledge-intensive, reasoning-driven T2I generation. These models excel at pixel-level detail synthesis, but their reliance on shallow text-image alignment limits them to surface-level semantic correlations.

Moreover, we evaluate the zero-shot performance of Qwen-Image by incorporating a CoT step through modifications to the system prompt. As shown in the results, this yields a slight improvement: the WISE score increases from 0.61 to 0.65. This is because of the lack of supervised signals to align its reasoning process with the need to infer visually actionable attributes suitable for DiT decoding.

To address this misalignment, we further conduct SFT training. However, the CoT remains unaware of the visual generation module DiT. This decoupling can lead to the LLM generating some strange tokens that the DiT cannot render into a reasonable image. This can be alleviated by further Dual-GRPO training, where the image-grounded rewards are used for LLM optimization, and DiT is adapted for better visual rendering. As seen, our models after SFT and Dual-GRPO achieves breakthrough performance on both benchmarks (0.79 on WISE, 68.3 accuracy on T2I-Reason). It achieves a leading performance across open-source T2I models and outperforms the leading proprietary model Gemini-2.0, approaching the SOTA of closed systems GPT-4o.

### 4.2.3. QUALITATIVE RESULTS

**Performance on real-world tasks.** In addition to reasoning-centric benchmarks, we evaluate our model's zero-shot performance on challenging real-world tasks, such as illustrat-

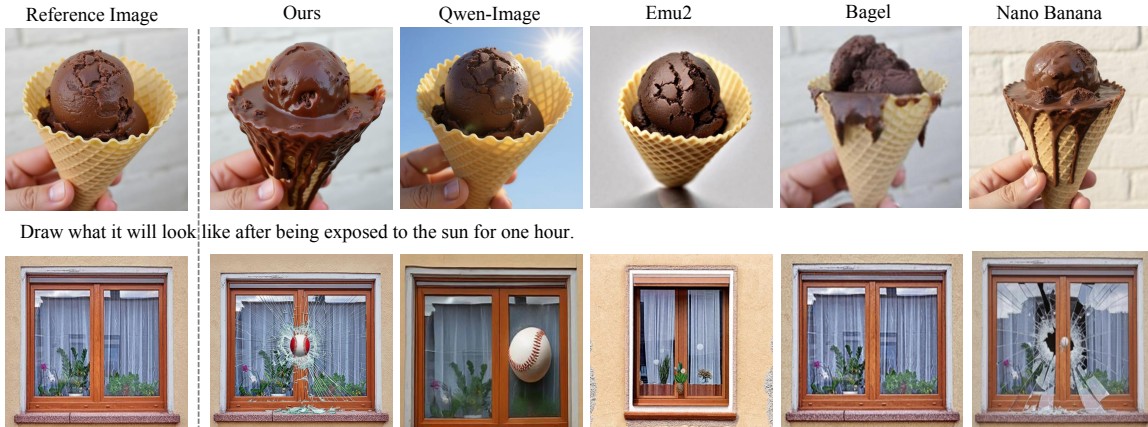

Reference Image · Ours · Qwen-Image · Emu2 · Bagel · Nano Banana

Draw what it will look like after being exposed to the sun for one hour.

Draw what it will look like after being hit by a baseball.

*Figure 5.* **Comparison of T2I models on conceptual image editing.** Vanilla Qwen-Image fails to interpret instructions (e.g., showing an ice cream under sunlight instead of melting), behaving as a text–pixel mapper. Our model correctly infers intended semantics, producing coherent, aesthetically pleasing edits with high consistency to the original image, outperforming unified models like Emu2 and Bagel.

*Table 2.* **Comparison of generation models on T2I-ReasonBench.** Numbers in bold are the highest score among open-source models.

| Model | Type | Idiom | | Textual | | Entity | | Scientific | | Overall ↑ | |
|---|---|---|---|---|---|---|---|---|---|---|---|
| | | Acc. | Qual. | Acc. | Qual. | Acc. | Qual. | Acc. | Qual. | Acc. | Qual. |
| FLUX.1-dev | *diffusion* | 39.1 | 83.4 | 56.9 | 76.5 | 45.1 | 90.6 | 46.7 | 80.9 | 47.0 | 82.8 |
| SD-3.5-medium | *diffusion* | 34.4 | 80.6 | 58.0 | 70.1 | 44.8 | 92.1 | 49.9 | 83.0 | 46.8 | 81.4 |
| SD-3.5-large | *diffusion* | 35.6 | 85.3 | 62.2 | 75.4 | 46.6 | 92.6 | 52.9 | 84.5 | 49.3 | 84.4 |
| Bagel w/CoT | *unified* | 44.6 | 84.3 | 44.0 | 73.7 | 52.4 | 91.6 | 57.7 | 88.3 | 49.7 | 84.5 |
| Janus-Pro-7B | *unified* | 25.5 | 78.0 | 37.2 | 70.9 | 38.5 | 87.6 | 44.9 | 77.8 | 36.5 | 78.6 |
| HunyuanImage-3.0 | *unified* | 25.4 | 80.2 | 54.2 | 80.9 | 52.3 | 92.2 | 56.8 | 84.4 | 47.2 | 84.4 |
| UniCoT | *unified* | 49.0 | 84.2 | 58.1 | 92.3 | 73.5 | 92.9 | 51.9 | 71.7 | 58.1 | 85.3 |
| Gemini-2.0 | *proprietary* | 52.4 | 87.8 | 73.0 | 83.3 | 67.0 | 94.3 | 66.7 | 89.3 | 64.8 | 88.7 |
| GPT-4o | *proprietary* | 75.7 | 94.5 | 86.9 | 97.6 | 77.5 | 96.6 | 74.7 | 94.3 | 78.7 | 95.8 |
| Qwen-Image | *diffusion* | 48.1 | 84.3 | 66.5 | 85.8 | 57.1 | 84.7 | 59.5 | 85.3 | 57.8 | 87.5 |
| Ours (w/o SFT&GRPO) | *diffusion* | 51.7 | 86.3 | 71.7 | 83.8 | 57.3 | 92.8 | 62.8 | 87.6 | 60.9 | 86.6 |
| Ours (w/o GRPO) | *diffusion* | 58.5 | 90.4 | 71.5 | 87.1 | 61.9 | 93.9 | 72.2 | 91.8 | 66.1 | 90.4 |
| Ours | *diffusion* | **58.5** | **90.6** | **74.2** | **89.5** | **68.8** | **95.2** | **71.7** | **93.5** | **68.3** | **92.2** |

ing math lessons. Results in Figure 2 demonstrate that our model generalizes well to these tasks, potentially assisting with daily human work. A user study on these tasks and more results can be found in Appendix D and F.

**Visualization of CoT reasoning during Dual-GRPO.** We visualize CoT reasoning process during Dual-GRPO in Figure 9. We observe instances where CoT fails to correctly infer raw user prompts in the rollout samples, leading to catastrophic outputs. These samples are assigned low rewards and penalized during GRPO, resulting in improved performance. Moreover, we identify cases where CoT correctly infers raw user prompts but still fails to generate the correct output images. This may arise from the misalignment between the latents of the revised prompt and the distribution preferred by DiT. These cases are also penalized during GRPO, highlighting the effectiveness of our image-aware reward design in adjusting textual reasoning. Moreover, the average length of CoT tokens on WISE is around 200, yielding low inference overhead.

### 4.3. Image Editing

#### 4.3.1. SUPERVISED FINE-TUNING

For raw user prompts for image editing, we select the UniREdit-Data-100K (Han et al., 2025), a large-scale synthetic training dataset specifically curated for image editing with CoT reasoning annotations. Following T2I tasks, we augment this dataset by instructing Gemini-2.5 (Google, 2025) to conclude a refined prompt for each raw prompt and corresponding CoT. Note that the original image is also provided to Gemini to generate the refined prompt. This augmented dataset is used to fine-tune Qwen-Image-edit. For Dual-GRPO training, we randomly select 5k raw user prompts from this dataset.

#### 4.3.2. QUANTITATIVE RESULTS

**Benchmarks.** We evaluate performance across various image editing benchmarks requiring reasoning capacity. UniREditBench (Han et al., 2025) evaluates models with

*Table 3.* **Comparisons of image editing results on UniREdit-Bench and RISEBench.** Numbers in bold indicate the highest score among open-source models. Underlined numbers denote the highest score. Detailed scores are shown in Appendix G.

| Model | Type | UniREdit ↑ | RISE ↑ |
|---|---|---|---|
| Gemini-2.5-Flash-Image | *proprietary* | 68.3 | 32.8 |
| GPT-Image-1 | *proprietary* | 73.4 | 28.9 |
| Seedream-4.0 | *proprietary* | 55.8 | 10.8 |
| Bagel w/ CoT | *unified* | 51.0 | 9.2 |
| UniWorld-V2 | *unified* | 54.9 | - |
| FLUX.1-Kontext | *diffusion* | 45.8 | 5.8 |
| Qwen-Image-edit | *diffusion* | 56.5 | 8.9 |
| Ours (w/o GRPO) | *diffusion* | 61.1 | 20.2 |
| Ours | *diffusion* | **68.7** | **23.9** |

2,700 meticulously curated samples, covering both real- and game-world scenarios. RISEBench (Zhao et al., 2025) comprises 327 prompts designed to evaluate models on reasoning-aware tasks across temporal, causal, spatial, and logical dimensions.

**Results analysis.** As shown in Table 3, after our post-training, Qwen-Image-edit achieves strong performance on image editing tasks, even surpassing Gemini-2.5-Flash-Image on the UniREdit benchmark. Moreover, it exhibits a substantial improvement over the SFT-only baseline, demonstrating the effectiveness of our Dual-GRPO in enhancing CoT reasoning through image-based rewards.

### 4.3.3. QUALITATIVE RESULTS

As shown in Figure 5, vanilla Qwen-Image-edit struggles to interpret conceptual editing instructions. For the prompt "draw what the ice cream looks like after being exposed to the sun," it merely renders the reference image under sunlight, resulting in visually incoherent output and revealing its behavior as a simple text–pixel mapper. In contrast, our model correctly infers the intended semantics, depicting the ice cream melting. More examples are shown in Appendix G.

### 4.4. Ablation Studies

To evaluate the effectiveness of SFT, we compare under two settings: (1) instructing vanilla Qwen2.5-VL to perform *T2G* without SFT, and (2) finetuning Qwen2.5-VL on the SFT dataset constructed by Qwen3-8B (Yang et al., 2025). Table 4 shows the model trained on Gemini-generated data achieves the highest upper bound, likely due to its higher-quality CoT and more reliable world knowledge, which produce more coherent refined prompts and provide DiT with stronger conditions. Notably, the model trained with the Qwen3-generated dataset and subsequently trained via GRPO achieves 0.76 on WISE, surpassing the purely Gemini-based SFT one, demonstrating the effectiveness of Dual-GRPO in fully enhancing models' reasoning capacity.

*Table 4.* Ablation studies on the importance of SFT. w/o SFT denotes the vanilla Qwen-Image with thinking system prompt.

| Model | WISE↑ | T2I-Reason↑ | |
|---|---|---|---|
| | | Acc. | Qual. |
| w/o SFT | 0.65 | 60.9 | 86.6 |
| w/o SFT + GRPO | 0.70 | 66.9 | 91.0 |
| w/ SFT (Qwen3-8B) | 0.71 | 65.0 | 90.0 |
| w/ SFT (Qwen3-8B) + GRPO | 0.76 | 67.8 | 91.6 |
| w/ SFT (Gemini2.5) | 0.74 | 66.1 | 90.8 |
| w/ SFT (Gemini2.5) + GRPO | **0.79** | **68.3** | **92.2** |

## 5. Related Work

**RL for diffusion models.** RL aligns generative models with human preferences, but faces challenges in DMs due to deterministic sampling and intractable likelihoods. Early approaches like DDPO (Black et al., 2023) discretized the reverse process for policy gradient updates but faced solver instability with complex prompts. Flow-GRPO (Liu et al., 2025) pioneered online RL integration for flow matching by converting ODEs to equivalent SDEs for exploration and optimizing only the DiT backbone. Similarly, Dance-GRPO (Xue et al., 2025) improved stability but also freeze the encoder. DiffusionNFT (Zheng et al., 2025) circumvented likelihood estimation by optimizing directly on the forward process via contrastive preference learning. In contrast, our Dual-GRPO framework co-optimizes both the LLM encoder and DiT in an end-to-end manner.

**Multimodal models for image generation** The shift toward unified multimodal architectures has accelerated with models such as HunyuanImage (Cao et al., 2025) and BAGEL (Deng et al., 2025), which integrate vision-language understanding and generation within a single autoregressive framework. While promising for complex prompts, these models remain biased toward literal generation due to reliance on descriptive caption data during pretraining. Recent post-training methods incorporate multimodal CoT reasoning to alleviate this issue, such as T2I-R1 (Jiang et al., 2025) and Uni-CoT (Qin et al., 2025). In contrast, our framework injects text-based CoT reasoning into the LLM encoder to enhance DMs.

## 6. Conclusion

This work addresses a key limitation of T2I DMs: their literal text-to-pixel mapping fails to leverage LLMs' reasoning and world knowledge. We propose the *think-then-generate* paradigm, transforming LLMs from passive encoders into active reasoning interpreters. We also propose Dual-GRPO, which jointly optimizes the LLM for image-grounded semantic consistency and the DiT for visual aesthetic. Our approach paves the way for next-generation models with enhanced reasoning capabilities.

## Impact Statement

This work focuses on a fundamental method for text-to-image diffusion models, with minimal ethical impact. While advancements in generative models could have societal consequences, the potential risks in this work are considered to be small.

## Acknowledgements

This work was supported by Shanghai Key Technology R&D Program "New Generation of Information Technology" (No. 25511103700), NSF of China (Nos. 62306176, 92470118), CCF-ALIMAMA TECH Kangaroo Fund (NO. CCF-ALIMAMA OF 2025010), Kuaishou Technology, and Ant Group.

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

| Prompt |
|---|
| You are a Prompt Optimizer specializing in image generation models (e.g., MidJourney, Stable Diffusion). Your core task is to rewrite user-provided prompts into highly clear, easy-to-render versions.
When rewriting, prioritize the following principles:
1. Start from the user's prompt, do reasoning step by step to analyze the object or scene they want to generate.
2. Focus on describing the final visual appearance of the scene. Clarify elements like the main subject's shape, color, and state.
3. If you are confident about what the user wants to generate, directly point it out in your explanation and the final revised prompt.
4. If technical concepts are necessary but difficult for ordinary users to understand, translate them into intuitive visual descriptions.
5. Ensure the final revised prompt is consistent with the user's intent.

After receiving the user's prompt that needs rewriting, first explain your reasoning for optimization. Then, output the final revised prompt in the fixed format of "Revised Prompt:\n". Where the specific revised content is filled in the next line.

Prompt: Refraction of light through water |

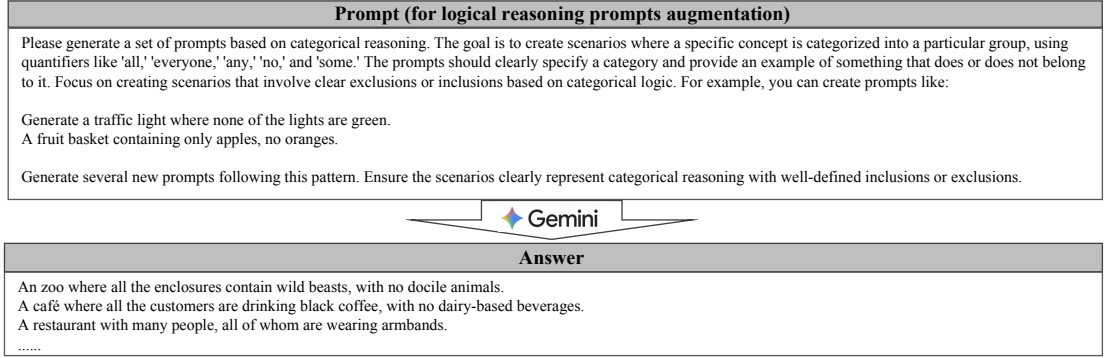

| Answer |
|---|
| The user wants to generate an image that visually represents the phenomenon of light refracting as it passes through water. This means focusing on how light bends and distorts objects seen through water, or how light itself appears distorted within the water.\n\nMy optimization reasoning is as follows:
1. **Identify the core concept:** "Refraction of light through water" is the key. This implies visual distortions, bending of light rays, and ...
2. **Focus on visual appearance:** How does refraction *look*? It looks like distortions, wavy patterns, shimmering effects, and the bending of straight lines.
3. **Clarify elements:**
  * **Main subject:** The interaction of light and water.
  * **Shape/Color/State:** The water should be clear enough for light to pass through, perhaps showing ripples or a surface. ...
4. **Translate technical to visual:...**
5. **Add descriptive details for clarity:...**
Revised Prompt:
A close-up, high-detail photograph of sunlight beaming through crystal clear water, creating mesmerizing optical distortions and wavy patterns on a submerged object. The light rays appear bent and fragmented as they pass through the water's surface, casting shimmering, ethereal refractions. Focus on the visual effects of light bending, with vibrant highlights and subtle caustics. |

*Figure 6.* An example of constructing the training data for the LLM encoder to replicate the T2G pattern.

| Prompt (for logical reasoning prompts augmentation) |
|---|
| Please generate a set of prompts based on categorical reasoning. The goal is to create scenarios where a specific concept is categorized into a particular group, using quantifiers like 'all,' 'everyone,' 'any,' 'no,' and 'some.' The prompts should clearly specify a category and provide an example of something that does or does not belong to it. Focus on creating scenarios that involve clear exclusions or inclusions based on categorical logic. For example, you can create prompts like:

Generate a traffic light where none of the lights are green.
A fruit basket containing only apples, no oranges.

Generate several new prompts following this pattern. Ensure the scenarios clearly represent categorical reasoning with well-defined inclusions or exclusions. |

| Answer |
|---|
| An zoo where all the enclosures contain wild beasts, with no docile animals.
A café where all the customers are drinking black coffee, with no dairy-based beverages.
A restaurant with many people, all of whom are wearing armbands.
...... |

*Figure 7.* An example of prompt set augmentation of the logical reasoning category of R2I-Bench.

## A. Details on Training Data Construction

We process the raw user prompt set to construct a supervised fine-tuning dataset using Gemini-2.5 (Google, 2025). This involves prompting Gemini-2.5 to perform CoT reasoning, where it descriptively infers the specific characteristics (such as color, texture, shape, and other details) that should be depicted, and then concludes a refined prompt. Detailed instructions for Gemini and an example of the answer are provided in Figure 6. For the image-editing task, the original image is also provided to Gemini as a reference.

## B. Details on Prompt Set Augmentation

R2I-Bench (Chen et al., 2025a) consists of 3,068 meticulously curated raw user prompts, spanning seven core reasoning categories: commonsense, logical, compositional, numerical, causal, and concept mixing. Each category is defined by specific criteria, such as: "Logical reasoning involves using systematic, structured approaches to analyze information, draw conclusions, and solve problems based on given premises or conditions.". We expand this prompt set to 5,000 by instructing Gemini-2.5 to generate additional examples based on these definitions and a set of prompt templates. An example of this curation process for the logical reasoning category is shown in Figure 7.

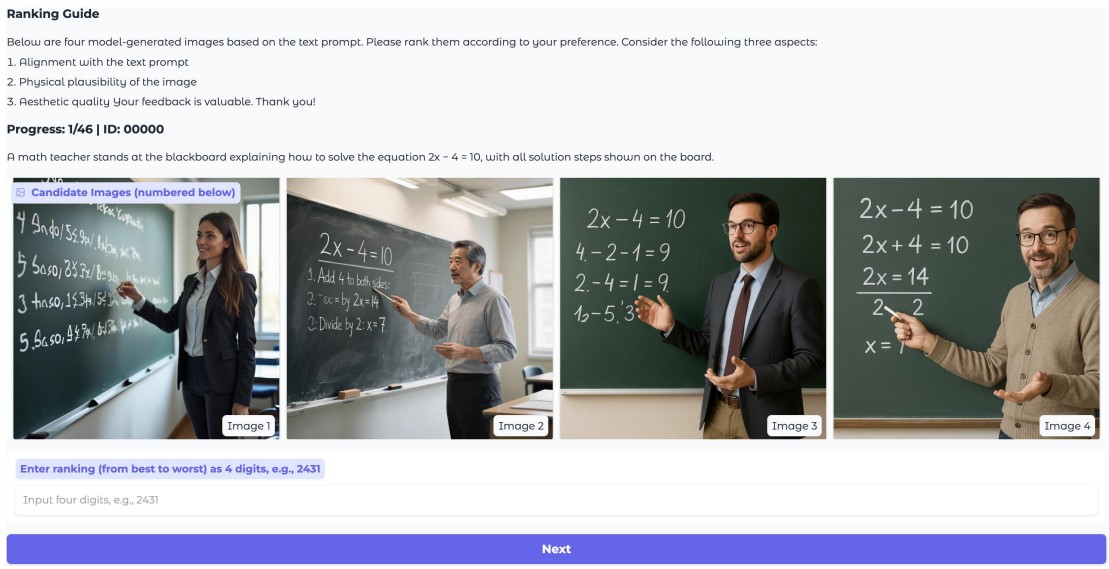

*Figure 8.* User interface for our human evaluation study.

*Table 5.* Comparison of different reward schedulers on T2I-ReasonBench.

| | Balanced Scheduler | | Staged Scheduler | |
|---|---|---|---|---|
| Category | Acc. | Qual. | Acc. | Qual. |
| Idiom | 58.5 | 90.6 | 59.5 | 90.6 |
| Textual | 74.2 | 89.5 | 73.5 | 88.5 |
| Entity | 68.8 | 95.2 | 66.9 | 94.8 |
| Scientific | 71.7 | 93.5 | 72.0 | 93.4 |
| Overall ↑ | 68.3 | 92.2 | 67.9 | 91.8 |

*Table 6.* Comparison of model performance on challenging tasks on user study. Count denotes the number of times the model receives each given rank.

| Count | Qwen-Image | Bagel | GPT-4o | Ours |
|---|---|---|---|---|
| rank 1 | 3 | 1 | 20 | 16 |
| rank 2 | 15 | 4 | 15 | 18 |
| rank 3 | 17 | 18 | 9 | 7 |
| rank 4 | 11 | 23 | 2 | 5 |
| Score ↑ | 2.21 | 1.63 | 3.15 | 2.98 |

## C. Ablation on Different Reward Schedulers

In Section 3.3, we introduce the reward-weighting scheduler used during training. Specifically, we adopt a balanced scheduler with constant weights, setting $\beta_1(\tau) = \beta_2(\tau) = 0.5$, such that both the LLM and the DiT are updated jointly at every iteration. In this section, we an alternative staged piecewise scheduler, where we set $\beta_1(\tau) = 1, \beta_2(\tau) = 0$ at early training steps and switch to $\beta_1(\tau) = 0, \beta_2(\tau) = 1$ in later steps (staged scheduler in Table 5. This design effectively updates only the LLM parameters in the early phase and only the DiT parameters in the later phase.

We compare the effectiveness of different schedulers in Table 5 on the T2I-ReasonBench benchmark. The results show that the balanced scheduler consistently outperforms the staged scheduler. We hypothesize that this is because joint optimization enables tighter coordination between prompt refinement and visual rendering.

## D. User Study

In this section, we conduct a user study on T2I generation in challenging real-world scenarios. We select 46 prompts relevant to meaningful tasks in everyday human life and compare our model against strong baselines, including vanilla Qwen-Image (Wu et al., 2025), Bagel-think (Deng et al., 2025), and GPT-4o (Hurst et al., 2024). Users are instructed to rank the model outputs according to *semantic alignment and conceptual understanding*, *visual realism and coherence*, and *aesthetic quality*. Our preference-collection interface is shown in Figure 8. For evaluation, we aggregate prompt-level rankings. For each prompt $i$, the model receives a reward $s_i = 5 - r_i$, where $r_i \in \{1, 2, 3, 4\}$ denotes its rank among the four models. The average score is the mean of all rewards (Table 6).

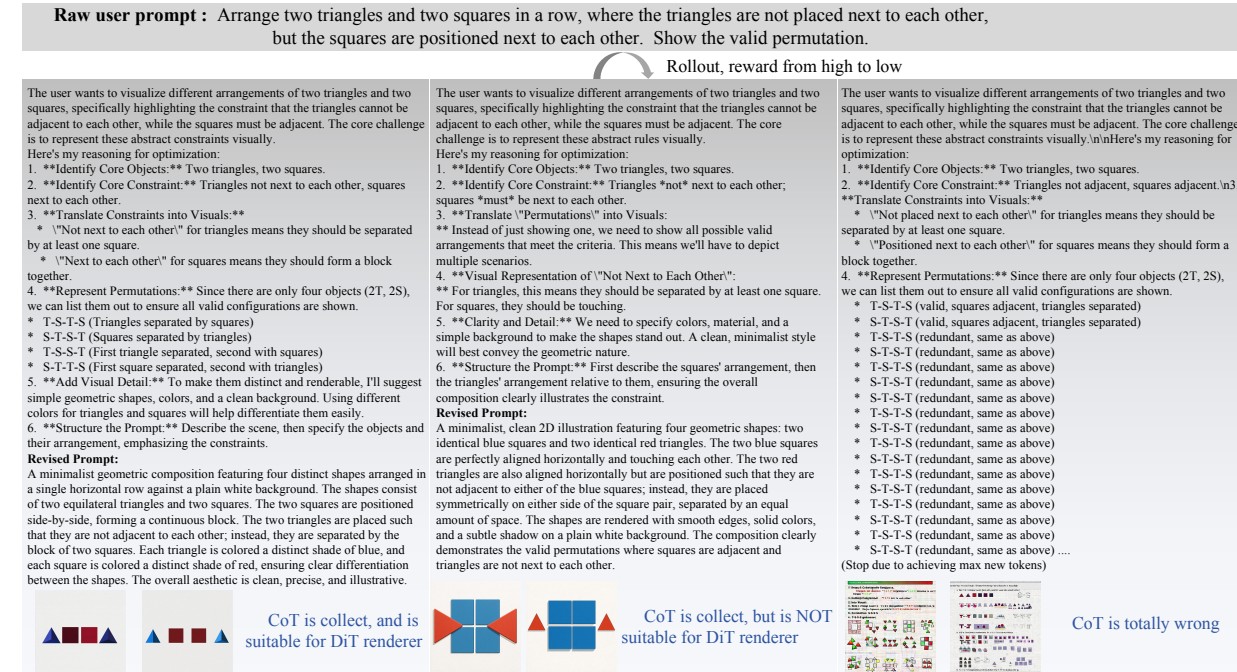

*Figure 9.* Visualization of CoT reasoning during Dual-GRPO.

*Table 7.* Detailed scores on RISEBench. Numbers in bold indicate the highest score among open-source models.

| Model | Type | Temporal | Causal | Spatial | Logical | Overall ↑ |
|---|---|---|---|---|---|---|
| Gemini-2.5-Flash-Image | *proprietary* | 25.9 | 47.8 | 37 | 18.8 | 32.8 |
| GPT-Image-1 | *proprietary* | 34.1 | 32.2 | 37 | 10.6 | 28.9 |
| Gemini-2.0-Flash-exp | *proprietary* | 8.2 | 15.5 | 23 | 4.7 | 13.3 |
| Seedream-4.0 | *proprietary* | 12.9 | 12.2 | 11 | 7.1 | 10.8 |
| BAGEL | *unified* | 2.4 | 5.6 | 14 | 1.2 | 6.1 |
| BAGEL (w/ CoT) | *unified* | 5.9 | 17.8 | 21 | 1.2 | 11.9 |
| FLUX.1-Kontext | *diffusion* | 2.3 | 5.5 | 13.0 | 1.2 | 5.8 |
| Qwen-Image-edit | *diffusion* | 4.7 | 10.0 | 17.0 | 2.4 | 8.9 |
| Ours (w/o GRPO) | *diffusion* | **22.3** | 27.8 | 21.0 | 9.4 | 20.2 |
| Ours | *diffusion* | 20.0 | **28.9** | **15.3** | **30.0** | **23.9** |

## E. Visualization of CoT Reasoning

In this section, we visualize the CoT reasoning and refined prompts within a single rollout. As shown in Figure 9, incorrect or imperfect CoT reasoning and their corresponding refined prompts are penalized during Dual-GRPO.

## F. More T2I Generation Results

We provide more demos for the T2I task in Figure 10. These results further demonstrate that our model delivers stronger conceptual instruction understanding and alignment, and improved overall visual plausibility and coherence compared to existing baselines.

## G. More Image-Editing Results

In this section, we provide detailed scores on the RISEBench (Zhao et al., 2025) in Table 7 and show additional qualitative results in Figure 11.

| Qwen-Image | Ours | FLUX.1 dev | GPT 4o | Nano Banana | Bagel |
|---|---|---|---|---|---|

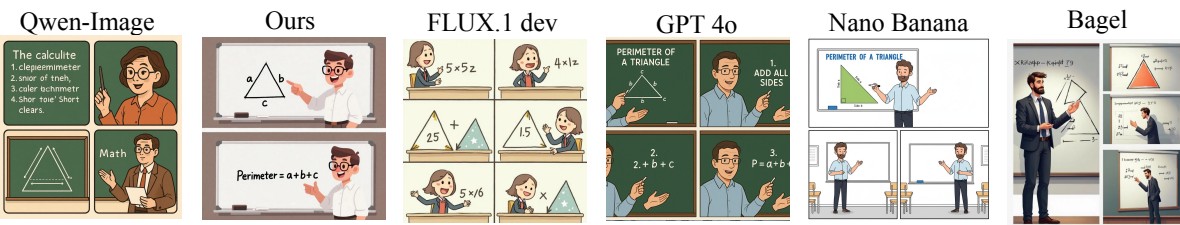

A multi-panel illustration of a math teacher explaining how to calculate the perimeter of a triangle.

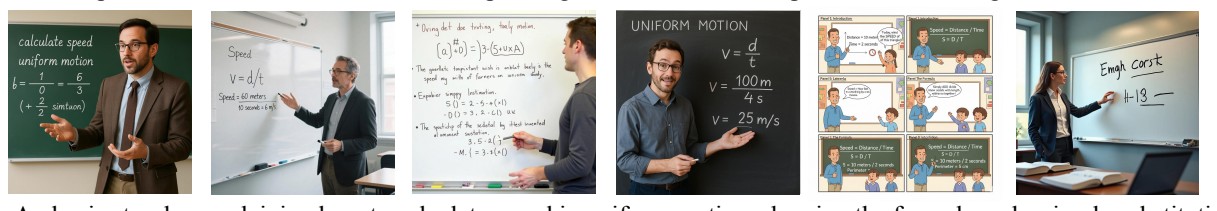

A physics teacher explaining how to calculate speed in uniform motion, showing the formula and a simple substitution.

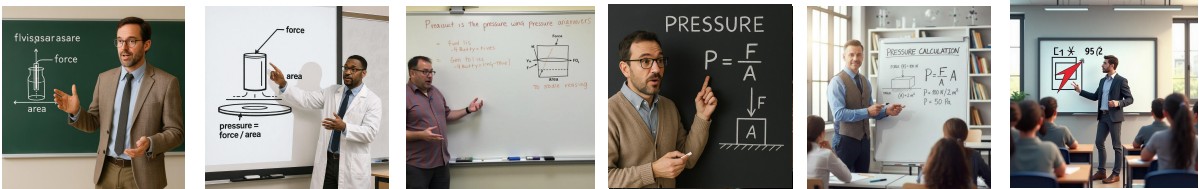

A physics teacher demonstrates how to calculate pressure using force and area, with a simple diagram.

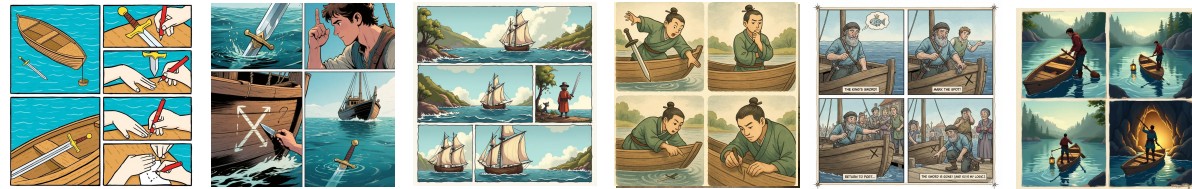

A multi-panel illustration showing the story of marking the boat to find a sword.

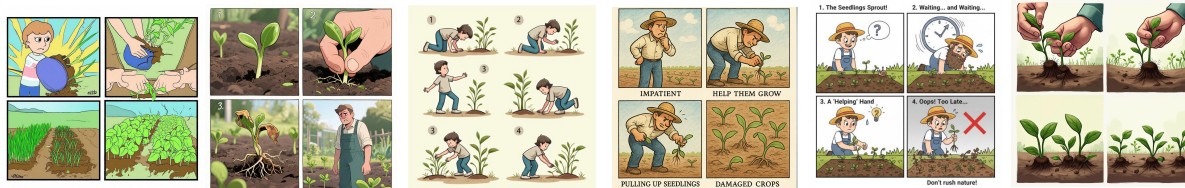

A multi-panel illustration showing the story of pulling up seedlings to help them grow, with clear steps from impatience to damaging the crops.

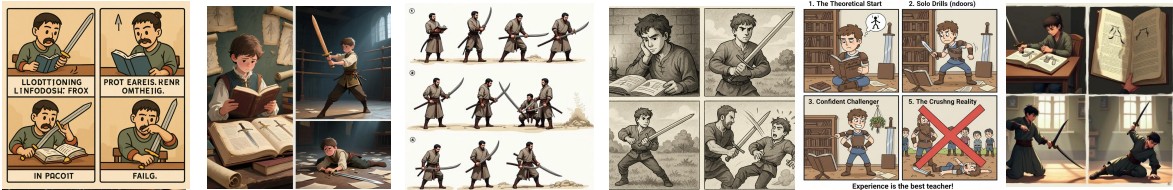

A multi-panel illustration showing the story of learning swordsmanship from a book only, with clear steps from reading theory to failing in practice.

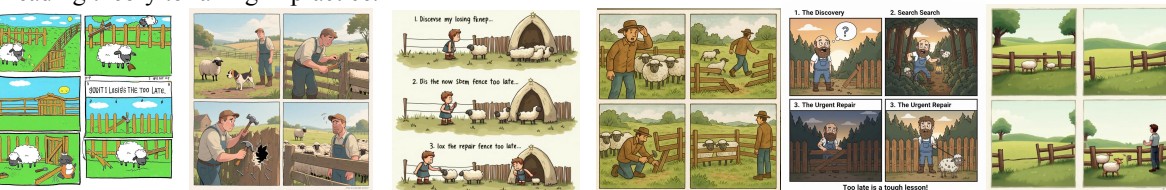

A multi-panel illustration showing the story of fixing the sheep pen after losing sheep, with clear steps from discovering the loss to repairing the fence too late

*Figure 10.* More demos for the T2I task.

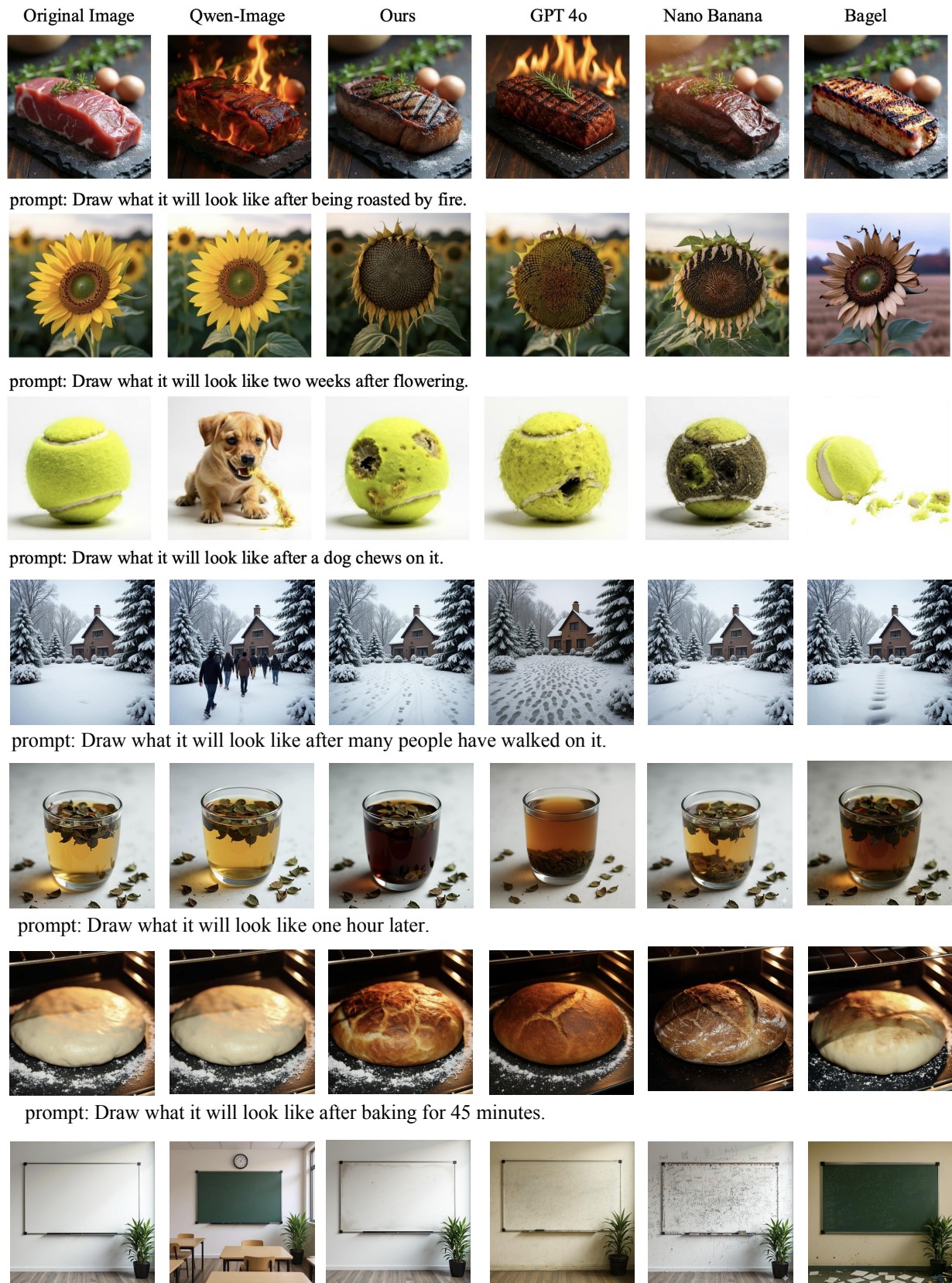

*Figure 11.* More demos for the image-editing task.

