# OpenReview forum: "Think-Then-Generate: Reasoning-Aware Text-to-Image Diffusion with LLM Encoders"
_ICML.cc/2026/Conference — ICML 2026 regular_

### Official Review · Reviewer_jK78 · 2026-02-23

**Soundness:** 3
**Presentation:** 3
**Significance:** 2
**Originality:** 2
**Overall Recommendation:** 4
**Confidence:** 3

**Summary:**

This paper proposes the Think-Then-Generate (T2G) paradigm for reasoning-aware text-to-image diffusion, aiming to move beyond treating large language models (LLMs) as frozen text encoders and instead activating their reasoning capabilities for visual generation. Concretely, the authors first fine-tune the LLM encoder with supervised data containing chain-of-thought reasoning and rewritten prompts, enabling a “think-then-rewrite” pattern that enriches raw user instructions. They then jointly optimize the LLM encoder and the diffusion backbone using a Dual-GRPO reinforcement strategy, where rewritten prompts bridge textual reasoning and image synthesis, and image-grounded rewards separately encourage semantic alignment in the encoder and visual quality in the diffusion model. Applied to Qwen-Image, the approach yields substantial improvements on multiple reasoning-oriented T2I and editing benchmarks.

**Compliance With Llm Reviewing Policy:**

Affirmed.

**Key Questions For Authors:**

1. Methods such as Uni-CoT and T2I-R1 already introduce reasoning into image generation. Could you more precisely articulate what is fundamentally new about T2G beyond training integration (e.g., is the main contribution Dual-GRPO, joint encoder–decoder RL, or something else)?

2. Since benchmarks such as WISE and T2I-ReasonBench rely on LLM/VLM-based judges, and your training also uses VLM-based rewards, how do you mitigate circularity or overfitting to the evaluator family?

3. Most experiments focus on reasoning-intensive benchmarks. Is there any degradation on non-reasoning prompts compared to the base model?

**Limitations:**

The paper does not include a dedicated limitations section.

**Strengths And Weaknesses:**

Strength:

--The paper clearly identifies a structural limitation of current diffusion-based T2I systems: LLMs are used as frozen encoders rather than reasoning agents. The think-then-generate (T2G) paradigm is logically derived from this diagnosis.

--Experiments cover text-to-image reasoning benchmarks (WISE, T2I-ReasonBench), image editing tasks (UniREdit, RISE), and ablations on SFT data sources and GRPO usage, demonstrating consistent and substantial improvements across all settings.

Weakness:

--Methods such as Uni-CoT and T2I-R1 also introduce reasoning for image generation. The novelty lies more in training integration than in conceptual paradigm shift.

--Benchmarks such as WISE and T2I-ReasonBench rely on LLM/VLM judges, raising potential circularity concerns.

--Although some rollout failures are discussed, systematic analysis of error cases is limited.

--Experiments focus primarily on reasoning-heavy prompts; broader evaluation on standard T2I quality/diversity benchmarks would clarify trade-offs.

---

> ### Author Rebuttal · Authors · 2026-03-31
>
> Thank you for the positive feedback and useful suggestions! We are encouraged that you recognize our paper 'clearly identified a structural limitation of current diffusion-based T2I systems and yielded substantial improvements on multiple benchmarks'. We address your concerns point by point below.
>
> **W1&Q1:** Methods such as Uni-CoT and T2I-R1 already introduce reasoning into image generation. Could you more precisely articulate what is fundamentally new about T2G beyond training integration (e.g., is the main contribution Dual-GRPO, joint encoder–decoder RL, or something else)?
>
> Thanks for this insightful question. Although Uni-CoT and T2I-R1 already introduce reasoning into image generation, **their focus is primarily on enhancing the capabilities of unified LMMs**. Specifically, T2I-R1 targets an autoregressive unified LMM Janus-pro, where RL is a direct extension of standard LLM GRPO. Uni-CoT constructs a multi-modal reasoning SFT dataset and applies it to fine-tune the unified LMM BAGEL.
>
> In contrast, **our work focuses on diffusion models with an LLM-DiT composite architecture, transforming the frozen text encoder into an active reasoning rewriter. To achieve this, we introduce Dual-GRPO, an RL framework that jointly optimizes the encoder (via GRPO for LLM) and decoder (via flow-GRPO for DiT) and achieves an optimal interplay between text reasoning and visual rendering.** This is non-trivial, as it requires careful design of the rollout trajectories across both components, the formulation of component-specific rewards, and a balanced joint optimization strategy (as ablated in Table 5 of our paper).
>
> Furthermore, our model trained under this framework achieves significantly higher reasoning-based generation performance than T2I-R1 and Uni-CoT (Table 1 and 2). We will include these clarifications in the revised version.
>
> **W2&Q2:** Since benchmarks such as WISE and T2I-ReasonBench rely on LLM/VLM-based judges, and your training also uses VLM-based rewards, how do you mitigate circularity or overfitting to the evaluator family?
>
> Thank you for this constructive feedback. **In Appendix D, we incorporate a user study evaluating challenging real-world text-to-image generation scenarios. Results in Table 6 show that our model strongly aligns with human preferences, substantially surpassing the baseline.** This human-centric evaluation confirms that our training indeed improves the model's generation capabilities, rather than merely overfitting the evaluator family.
>
> **W3:** Although some rollout failures are discussed, systematic analysis of error cases is limited.
>
> We appreciate this constructive feedback. To address this, we filter 50 rollouts with a semantic score of 0 and manually annotate these failed generation samples. Our analysis reveals three failure modes: (1) LLM reasoning failures (68%), where the CoT is incorrect; (2) DiT incompatibility (30%), where CoT is accurate, but the refined prompt is not suitable for DiT render; and (3) Judging hallucination (2%), where the VLM judge hallucinates and misjudges the image. We will add this quantitative analysis to the revised Appendix.
>
> **W4&Q3:**  Is there any degradation on non-reasoning prompts compared to the base model?
>
> Thanks for this constructive suggestion. To address this, we further evaluate our model on DPG-Bench[1], a standard T2I benchmark comprising 1k non-reasoning prompts, and HPSv2[2], a T2I benchmark testing the model on a total of 3200 prompts, with 800 prompts for each of the following styles: “animation”, “concept-art”, “painting”, and “photo”. The results are detailed in the table below.  As seen, **there is no degradation in non-reasoning prompts, and an improvement is observed due to the descriptive refinement of the original prompts**. We will add this result to the revised version.
>
> | | DPG-Bench$\uparrow$ | HPSv2$\uparrow$ |
> | :--- | :--- | :--- |
> | Qwen-Image | 88.32 | 28.28 |
> | Ours | **88.53** | **28.49** |
>
> [1] ELLA: Equip Diffusion Models with LLM for Enhanced Semantic Alignment. [https://arxiv.org/abs/2403.05135](https://arxiv.org/abs/2403.05135)
>
> [2] Human Preference Score v2: A Solid Benchmark for Evaluating Human Preferences of Text-to-Image Synthesis. [https://arxiv.org/abs/2306.09341](https://arxiv.org/abs/2306.09341)
>
> Given these clarifications and improvements to your initial concerns, we hope for your reconsideration in raising your score.

---

> > ### Author Rebuttal · Reviewer_jK78 · 2026-04-03
> >
> > Thank authors for the detailed response. I would like to maintain my original positive rating.

---

> > > ### Author Response · Authors · 2026-04-03
> > >
> > > Thanks again for your positive feedback and constructive suggestions, which have been very helpful in improving this work.

---

### Official Review · Reviewer_HNGt · 2026-03-11

**Soundness:** 3
**Presentation:** 2
**Significance:** 3
**Originality:** 2
**Overall Recommendation:** 4
**Confidence:** 3

**Summary:**

This paper presents an interesting idea for text-to-image generation that instead of treating the LLM encoder as a frozen part, it turns the encoder into a rewriter.
The proposed T2G framework first constructs SFT data with "raw user prompts, CoT reasoning, rewritten prompts", and then fine-tunes the LLM.
Then, T2G jointly optimize the LLM text encoder and the DiT generator under the framework of Dual-GRPO with tailored rewards.
The LLM text encoder is encourgaed to improve semantic alignment and consistency, while the DiT is optimized for visual realisom and aesthetic quality.
Overall, the reported results show that T2G improves performance on conceptual editing instrcution and reasoning-required T2I task.

**Compliance With Llm Reviewing Policy:**

Affirmed.

**Final Justification:**

The authors have addressed all my concerns on the motivation of physical reward and CoT inference overheating during the discussion period, and my final rating has been raised to 4 now.

**Key Questions For Authors:**

1. Address the above weaknesses. Focusing on the explanation of $R_phy$ definition and provide the reward details. I will increase my score once the author could address my concern.
2. While the proposed Think-Then-Generate framework also introduces a reasoning phase through CoT-based prompt rewriting, it is not yet entirely clear what fundamental differences the proposed approach has from these recent reasoning-based T2I frameworks (T2I-R1 [1]). Could the author provide a clear explanation for the differences?
3. Correct typos and function parameters definition, like $s_t$ in equation (5).

[1] Jiang, Dongzhi, et al. "T2i-r1: Reinforcing image generation with collaborative semantic-level and token-level cot." NeurIPS 2025.

**Limitations:**

yes

**Strengths And Weaknesses:**

Strengths:
1. The proposed think-then-generate framework introduces the reasoning step before image generation is interesting.
They leverage an LLM-based text encoder to perform prompt rewriting with CoT reasoning, which improve the semantic and aesthetic quality compared to the baseline model and provide a promising step for reasoning-awared generation.
Overall, the proposed method shows improvements over the baseline Qwen-Image across multiple image generation and editing benchmarks.
2. Training pipeline is clear and well-structured, including SFT dataset construction, SFT, and Dual-GRPO training.

Weaknesses:
1. The motivation for incorporating the physical reward $R_{phy}$ with the aesthetic reward for DiT training is not sufficiently claimed.
Furthermore, the manuscirpt lack technical details for the design of these rewards in Dual-GRPO, like the reward function of $R_{phy}$, the $\omega_1$ and $\omega_2$ setting.
Notably, while Equation 12 defines $R_2$ as a combination of $R_{aes}$ and $R_{phy}$, Figure 4 only presents training accuracy for the "Semantic" and "Aesthetic" components, omitting the physical aspect, and this visualization (Figure4 (b)) is ambiguous.
2. Since the proposed framework introduces the reasoning step before image generation, it likely increased inference latency compared to standard text encoders. However, the paper lacks the discussion of computational cost introduced by this CoT process.
In addition, Table 3 shows that the method still underperforms GPT-Image-1 on both UniREdit and RISE bench, and only slightly outperforms Gemini-2.5-Flash-Image on UniREditBench of 0.4, while being significantly lower on RISEBench. Given these results, it would be valuable for the authors to report inference time comparisons with baseline models and state-of-the-art systems to better assess the trade-off between performance gains and computational cost.
3. Some typos. The paper claims that the proposed method achieves a result of 73.4 on the UniREditBench, while Table 3 reports a score of 68.7 for the proposed model, with 73.4 corresponding to GPT-Image-1. This discrepancy should be clarified and corrected to avoid confusion.

---

> ### Author Rebuttal · Authors · 2026-03-31
>
> Thank you for your thoughtful review and valuable feedback! We are encouraged that you find our approach to turning the frozen LLM encoder into an active rewriter interesting. We address your concerns point by point below.
>
> **W1:** Lack of motivation and explanation of  $R_{phy}$ and reward details.
>
> We thank the reviewer for the valuable suggestions and apologize for the initial confusion.
>
> First, regarding the motivation for $R_{phy}$, DiT rendering occasionally breaks fundamental laws of physics (e.g., incorrect shadow casting). To explicitly enforce visual realism, we incorporate $R_{phy}$ alongside the aesthetic reward. In practice, we balance this with the aesthetic reward by setting $w_1=0.5$ and $w_2=0.5$. We will clarify this in the revised version.
>
> Second, to provide complete details on the reward design, **we provide the full system prompts for the VLM judge used to evaluate semantic consistency, aesthetic quality, and physical consistency** on an anonymous supplementary page due to the character limit: [https://pagedrop.dev/s/uFKALBUC/](https://pagedrop.dev/s/uFKALBUC/). For immediate reference, we highlight the core criteria from the system prompt used to assess physical consistency below:
> ```text
> **Realism (0-1):** How realistically the image is rendered.
> * **0 (Rejected):** Physically implausible and clearly artificial. Breaks fundamental laws of physics or visual realism.
> * **0.5 (Conditional):** Contains minor inconsistencies or unrealistic elements. While somewhat believable, noticeable flaws detract from realism.
> * **1 (Exemplary):** Achieves photorealistic quality, indistinguishable from a real photograph. Flawless adherence to physical laws, accurate material representation, and coherent spatial relationships. No visual cues betraying AI generation.
> ```
> We will include these details in the Appendix in the revised version.
>
> Finally, we apologize for the ambiguity in Figure 4(b). We originally omitted the $R_{phy}$ curve due to spatial constraints in the main text. In practice, $R_{phy}$ reward also shows a stable upward trend during training. We will add the training curve in the revised Appendix.
>
> **W2:** Inference time comparisons with baseline models and state-of-the-art systems to better assess the trade-off between performance gains and computational cost.
>
> Thank you for pointing out this practical concern. We compare the average inference time of our method on the WISE benchmark with baselines and state-of-the-art commercial systems. As shown in the table below, **the CoT generation requires only about 6s on a single A100 GPU. Given that the standard DiT denoising process (50 steps) takes 50s, the latency overhead introduced by the CoT process is low.** We will add this to the revised version.
>
> | Model | Hardware / Platform | Inference Time (seconds per image) |
> | :--- | :--- | :--- |
> | Qwen-Image | A100-GPU | 50 |
> | Qwen-Image w/ T2G | A100-GPU | 56 (50s DiT + 6s CoT) |
> | Gemini-2.5-Flash-Image | API | 20 |
>
> **W3&Q3:** Correct typos and function parameters definition
>
> We sincerely thank the reviewer for the careful review and for pointing out these typos. To clarify Equation 5, $s_t$ represents the current state: it is $(z_{<t}, q)$ during the text CoT generation ($t \le l$) and $(x_{t-1}, \hat{z})$ during the DiT rendering ($t > l$). We will update the manuscript to clearly state this definition and correct the typo regarding the UniREditBench score.
>
> **Q2:** What fundamental differences does the proposed approach have from these recent reasoning-based T2I frameworks, such as T2I-R1?
>
> Thank you for this insightful feedback. While we share the conceptual goal of introducing a reasoning phase before image generation with recent works like T2I-R1 [1], the fundamental differences lie in model architectures and the corresponding RL framework.
>
> **T2I-R1 focuses on autoregressive unified LMM (i.e., Janus-Pro) where both text CoT and images are represented as discrete tokens.** Consequently, their RL process is a standard extension of standard GRPO for LLMs. In contrast, **our work focuses on diffusion models with an LLM-DiT composite architecture.** Specifically, Dual-GRPO jointly optimizes the encoder (via GRPO for LLM) and decoder (via flow-GRPO for DiT) to achieve an optimal interplay between text reasoning and visual rendering. This is non-trivial, as it requires careful design of the rollout trajectories across both components, the formulation of component-specific rewards, and a balanced joint optimization strategy (as ablated in Table 5 of our paper).
>
> Furthermore, our model achieves significantly higher reasoning-centric generation performance, outperforming T2I-R1 on the WISE benchmark (0.79 vs. 0.54 in Table 1). We will include these clarifications in the revised version.
>
> We hope these clarifications and explanations of our method in response to your initial concerns can convince you to increase your score.

---

> > ### Author Rebuttal · Reviewer_HNGt · 2026-04-03
> >
> > Thanks for your response. My concerns about the motivation of physical reward and CoT inference overhead have been addressed. I will adjust my rating.

---

> > > ### Author Response · Authors · 2026-04-08
> > >
> > > Thank you for the kind acknowledgment that your concerns were addressed. We really appreciate your feedback! In light of the fast approaching of final justification, we would like to send a gentle reminder regarding the rating adjustment you kindly mentioned in your previous comment. Thanks again for your time, effort, and support for our work!

---

### Official Review · Reviewer_HToE · 2026-03-12

**Soundness:** 3
**Presentation:** 2
**Significance:** 2
**Originality:** 2
**Overall Recommendation:** 4
**Confidence:** 4

**Summary:**

This paper introduces Think-Then-Generate, a framework to leverage LLM's own reasoning and prompt rewriting capablities to enable world-knowledge based text-to-image generation. With Dual-GRPO, this method surpasses baselines in world knowledge  T2I benchmarks.

**Compliance With Llm Reviewing Policy:**

Affirmed.

**Final Justification:**

The rebuttal has resolved my concerns.

**Key Questions For Authors:**

1. How much inference cost is introduced?

**Limitations:**

Scaling behaviour and the inference cost of this method.

**Strengths And Weaknesses:**

Strengths:
1. This paper is well-motivated and easy to follow.
2. The method is generally straightforward and simple to use.
3. The experimental results are convincing.

Weakness:
1. The overall framework's novelty is limited. This paper mainly combines LLM prompt-rewrite and GRPO for two processes. Both methods are largely explored in previous methods[1,2]. The dual-grpo mainly combines two rewards into one training procedure.
2. The author claims that the changed embeddings are suitable for T2I models because the t-SNE maps are not too different. So why not train a simple MLP to further align with original maps? Given the author already conducted GRPO with much training cost, this simple MLP should be very efficient and cost-friendly.

3. The reward models depend heavily on LLM/VLM judges.
Is Qwen3-30B-A3B the best judge? What about using other advanced VLM models?

4. Experimental comparisons should be made against other post-training methods, not the baselines. For example, Qwen-Image + CoT via system prompt, or simple RL baselines Flow-GRPO or Dance-GRPO.



[1] Wang, Linqing, et al. "Promptenhancer: A simple approach to enhance text-to-image models via chain-of-thought prompt rewriting." arXiv preprint arXiv:2509.04545 (2025).
[2] Geng, Zigang, et al. "X-omni: Reinforcement learning makes discrete autoregressive image generative models great again." arXiv preprint arXiv:2507.22058 (2025).

---

> ### Author Rebuttal · Authors · 2026-03-31
>
> Thanks for your detailed reviews. We are glad that you think our paper is well-motivated and our method is simple to use. We address your concerns point by point below.
>
> **W1:** This paper mainly combines LLM prompt-rewrite and GRPO for two processes. Both methods are largely explored in previous methods[1,2].
>
> **We want to clarify that Dual-GRPO is a novel framework extending GRPO to composite LLM-DiT models. It jointly optimizes the encoder (via GRPO for LLM) and decoder (via flow-GRPO for DiT) and achieves an optimal interplay between text reasoning and visual rendering.** This is also acknowledged by Reviewer qagR that our Dual-GRPO framework correctly extends GRPO to composite LLM-DiT models, with clear mathematical formulations for both components.
>
> Moreover, there are significant differences compared to two previous methods:
> - **PromptEnhancer[1] utilizes an external LLM as a prompt rewriting tool** and optimizes it using image-based rewards from a frozen DiT. This ignores the diffusion model's internal LLM encoder, failing to unlock the native reasoning capacity of the model itself. Moreover, **keeping the DiT frozen limits the model's upper bound.** To explicitly validate this, we conduct an ablation study where we freeze the DiT and solely optimize the LLM (β1=1, β2=0) and compare results on T2I-ReasonBench. As shown below, our joint optimization strategy (β1=0.5, β2=0.5) achieves the optimal performance.
> | | Acc $\uparrow$ | Qual $\uparrow$ |
> | :- | :- | :- |
> |β1=1, β2=0|66.8|90.4|
> |β1=0.5, β2=0.5|69.8 | 92.6 |
>
> - **The architecture and RL framework of X-omni[2] differ fundamentally from our work. X-omni is an autoregressive image generation model**; once the LLM generates the visual semantic tokens, the subsequent image decoding process is deterministic (a one-to-one mapping). Thus, the visual detokenizer does not require RL optimization. In contrast, **our focus is on diffusion models with LLM-DiT composite architecture.** Even when the text embeddings are fixed, the diffusion sampling process remains inherently stochastic, producing diverse images (a one-to-many mapping). Consequently, incorporating flow-GRPO into the DiT is necessary to guide the denoising trajectory toward visual alignment and aesthetic quality.
>
> We are sorry for the confusion and will add discussions of both [1] and [2] in the revised version to highlight these differences.
>
> **W2:** Why not train an MLP to further align with original maps？
>
> Our t-SNE results confirm SFT natively preserves latent structure, ensuring DiT compatibility. We further evaluate MJHQ-30k[1] to support this claim: **the post-SFT Qwen2.5-VL maintains an FID of 7.84, showing almost no degradation compared to the original model's 7.82.** Moreover, direct alignment of high-dimensional embeddings is empirically difficult. To investigate this, we train an MLP with MSE loss to map post-SFT embeddings back to the pre-SFT space. This deterministic alignment causes representation collapse, worsening the FID to 8.34. We will add these to the revised version.
>
> [1] Playground v2.5: Three Insights towards Enhancing Aesthetic Quality in Text-to-Image Generation.
>
> **W3:** Is Qwen3-30B-A3B the best judge? What about using other advanced VLM models?
>
> To address this, we randomly sample 100 prompt-image pairs from our RL rollouts and evaluate using advanced VLMs. **Qwen3-30B-A3B shows high Pearson correlations with advanced VLMs for semantic/aesthetic rewards**: 0.92/0.95 against Qwen3-235B-A22B, and 0.89/0.87 against Gemini-2.5-Flash. It delivers robust signals while remaining cost-effective.
>
> **W4:** Comparisons against other post-training methods. For example, Qwen-Image + CoT via system prompt, or simple RL baselines Flow-GRPO.
>
> **We would like to clarify that our manuscript has already included comparisons with the Qwen-Image + CoT via system prompt.**  Specifically, we evaluate the zero-shot performance of Qwen-Image by incorporating a CoT step through system prompt (L288 and L361). Furthermore, we also evaluate the model post-trained via SFT using Gemini-generated CoT (L289 and L362).
>
> Regarding simple RL baselines, we additionally train the DiT backbone using vanilla Flow-GRPO. This experiment utilizes the same training prompt set and a balanced 1:1 semantic-aesthetic reward, but excludes CoT process. **Results below show that this baseline falls far behind our Dual-GRPO framework.** We will add this to the revised version.
>
> | | WISE$\uparrow$ | T2I-Reason$\uparrow$ |
> |:-|:-|:-|
> |Qwen-Image|0.61|57.8|
> |Flow-GRPO|0.64|60.1|
> |Ours|**0.79**|**68.3**|
>
> **Q1:** How much inference cost is introduced?
>
> We appreciate this practical point. On the WISE benchmark, CoT generation (~200 tokens) takes ~6s on a single A100 GPU, adding minimal overhead compared to the 50s required for the DiT denoising process.
>
> Given these reclarifications of our method and additional experiments in response to your initial concerns, we hope for your reconsideration in raising your score.

---

> > ### Author Rebuttal · Reviewer_HToE · 2026-04-03
> >
> > Thanks for your efforts. However, my concerns are not fully addressed.
> > 1. I understand the authors’ argument that Dual-GRPO extends GRPO to a composite LLM-DiT architecture and jointly optimizes both components with different reward signals. But I still find the method to be more incremental and integrative.
> >
> >     Meanwhile, I also noticed the $\beta_1, \beta_2 $ configuration. This seems to be a default setting and does not have a closer analysis. The ablation in Table 5 indicates a relatively small advantage over the staged scheduler. Can the author provide results on other benchmarks? It seems that using text grpo first and flow-grpo second still yields great results, with only small changes in quality, and 2/4 improvements across categories over Dual-GRPO.
> >
> > 2. About comparisons for other post-training methods. What about Uni-CoT or T2I-R1, apart from vanilla experiments?
> >
> > 3. I appreciate the author's efforts in providing additional experiments for t-SNE and VLM judgment. I strongly encourage the authors to incorporate them in the revised manuscript.
> >
> > Overall, the rebuttal has addressed some of the concerns, and I will increase my score to 3 for now.

---

> > > ### Author Response · Authors · 2026-04-05
> > >
> > > Thanks for your follow-up feedback! We address your concerns point by point below.
> > >
> > > **C1:** I understand that Dual-GRPO extends GRPO to a composite LLM-DiT architecture and jointly optimizes both components with different reward signals, but the method seems integrative. Meanwhile, the staged scheduler yields similar results to the joint optimization. Can the author provide results on other benchmarks?
> > >
> > > We would like to clarify that **a naive integration of GRPO into an LLM-DiT architecture leads to suboptimal performance**.
> > > - First, **a naive staged optimization—training the LLM first and then the DiT—also results in suboptimal performance.** To further validate this, **we expand our evaluation on the WISE benchmark. The results show that our balanced joint scheduler achieves a score of 0.79, outperforming the staged scheduler (0.76).** This confirms that joint optimization is more effective, as it enables tighter coordination between prompt refinement and visual rendering.
> > >
> > > - Second, **collecting rollout trajectories via a naive sampling strategy where each sample pairs a reasoning CoT with one corresponding image (i.e., $K=1$ in Figure 4(a)) is suboptimal.** To demonstrate this, we conduct experiments on this single-branch strategy, comparing it against our hierarchical tree-structured rollout strategy. As shown in the table below, performance degrades significantly under this strategy. This is because it ignores the inherent stochasticity of the denoising trajectory induced by the DiT, which yields an **unreliable reward estimation** for the reasoning-conditioned image generation distribution.
> > > | | WISE $\uparrow$| T2I-Reason (Acc.) $\uparrow$ | T2I-Reason (Qual.) $\uparrow$ |
> > > | :--- | :--- | :--- | :--- |
> > > | single-branch | 0.75 | 66.5 | 90.1 |
> > > | hierarchical (ours) | **0.79** | **68.3** | **92.6** |
> > >
> > > Overall, extending GRPO to a composite LLM-DiT architecture is non-trivial. **It necessitates the careful design of rollout strategy to assign suitable credit for LLM reasoning and DiT rendering, alongside a balanced joint optimization strategy.** To the best of our knowledge, our work is the first to achieve this. Thanks for this further feedback, and we will add these ablations to further solidify our contributions in our revised version.
> > >
> > > **C2:** Comparisons for other post-training methods Uni-CoT or T2I-R1.
> > >
> > > **We would like to first clarify that we have already included comparisons with both T2I-R1 and Uni-CoT in the current manuscript (see Tables 1 and 2).** The results demonstrate that both methods lag behind our method: T2I-R1 achieves a score of 0.54 and Uni-CoT achieves 0.75, compared to our 0.79 on the WISE benchmark.
> > >
> > > Moreover, **both Uni-CoT and T2I-R1 are post-training methods specifically designed to enhance unified LMMs. Thus, directly applying them to diffusion models is infeasible.** Specifically, Uni-CoT constructs a multi-modal text-image interleaved reasoning SFT dataset and applies it to fine-tune the unified LMM BAGEL. Because Qwen-Image does not support interleaved text-image generation, training this dataset on it is incompatible.
> > >
> > > Regarding T2I-R1, it targets Janus-pro, an autoregressive unified LMM. Because Janus-Pro represents both text and images as discrete tokens, its RL approach simply extends standard GRPO. **Adapting this methodology to our framework is conceptually analogous to optimizing the discrete tokens generated from the LLM** (i.e., β1(τ) = 1, β2(τ) = 0). As we have shown in the previous rebuttal, **solely optimizing the LLM exhibits a limited performance upper bound** (66.8 on the T2I-ReasonBench), whereas our joint optimization strategy achieves the optimal performance with a score of 69.8.
> > >
> > > We welcome further discussion and kindly ask for reconsideration in raising your score if these clarifications and additional experiments fully address your concerns.

---

### Official Review · Reviewer_qagR · 2026-03-18

**Soundness:** 3
**Presentation:** 3
**Significance:** 3
**Originality:** 4
**Overall Recommendation:** 5
**Confidence:** 5

**Summary:**

This paper addresses a fundamental limitation in current text-to-image diffusion models: their inability to leverage the reasoning capabilities and world knowledge of large language models, resulting in literal 'text-to-pixel' mapping that fails on prompts requiring conceptual understanding. The authors propose a 'think-then-generate' paradigm where the LLM encoder first performs chain-of-thought reasoning to interpret user intent and generate detailed visual descriptions before passing representations to the diffusion transformer for image generation. The paper makes two main contributions: (1) a supervised fine-tuning approach to activate reasoning-aware behavior in the LLM encoder using Gemini-2.5-generated CoT data, and (2) Dual-GRPO, a reinforcement learning strategy that jointly optimizes both the LLM encoder and DiT using image-based rewards—semantic consistency for the encoder and visual aesthetics for the generator. Experiments on text-to-image and image editing tasks demonstrate improvements over baselines, with ablation studies confirming the effectiveness of both SFT and GRPO components.

**Compliance With Llm Reviewing Policy:**

Affirmed.

**Key Questions For Authors:**

1) How did you validate the reliability of Qwen3-30B-A3B as a judge for semantic consistency and aesthetic quality?
2) How does Dual-GRPO compare to alternative approaches like (a) first optimizing the LLM with text-based rewards, then the DiT, or (b) using a single reward function for both components?
3) The paper focuses on Qwen-Image and MM-DiT. How transferable is your approach to other LLM-DiT architectures (e.g., PixArt-α with different text encoders, or FLUX with T5 encoders)? Have you conducted any experiments with alternative architectures to assess generalizability?

**Limitations:**

Yes

**Strengths And Weaknesses:**

The proposed Dual-GRPO framework correctly extends GRPO to composite LLM-DiT models, with clear mathematical formulations for both components. The experimental design is appropriate, with evaluations on established benchmarks (WISE, R2I-Bench, T2I-ReasonBench, UniREDitBench, RISEBench) and comparisons against both open-source and proprietary models. The ablation studies systematically validate each component's contribution. The empirical gains on benchmarks like WISE (0.79 vs. 0.65 baseline) are substantial and demonstrate real practical utility. Speaking about the insight to use different reward signals for the two components (semantic consistency for LLM, aesthetics for DiT) is clever and well-justified by their distinct roles.

However, there are some concerns:
1) The reward functions rely heavily on a VLM-as-a-judge approach using Qwen3-30B-A3B, but the paper lacks analysis of the judge model's reliability or potential biases
2) The scheduler with constant weights (β₁=β₂=0.5) is mentioned but deferred to Appendix C for ablation, making it difficult to assess whether this choice is optimal
3) The authors claim that SFT preserves latent space structure but only provide a qualitative t-SNE visualization without quantitative metrics to support this claim
4) The scope of impact may be somewhat limited by the reliance on specific model architectures (Qwen-Image/MM-DiT), though the general principles should transfer to other LLM-DiT composite models
5) The paper lacks an expected discussion of computational costs and training efficiency

---

> ### Author Rebuttal · Authors · 2026-03-31
>
> Thank you for the positive feedback and useful suggestions! We are glad you thought our paper 'addressed a fundamental limitation in current text-to-image diffusion models'. We address your concerns point by point below.
>
> **W1 & Q1:** The reliability of Qwen3-30B-A3B as a judge. How did you validate the reliability of Qwen3-30B-A3B as a judge for semantic consistency and aesthetic quality?
>
> Thanks for this constructive suggestion. To address this concern, we evaluate the agreement between the rewards assigned by Qwen3-30B-A3B and those from human experts using a random sample of 100 prompt-image pairs from our RL rollouts. As shown in the table below, **the high Pearson correlation metrics between Qwen3-30B-A3B and human experts demonstrate robust consistency in judging semantic consistency and aesthetic quality.** We will include this in the revised manuscript.
>
> | semantic consistency | aesthetic quality |
> | :--- | :--- |
> | 0.81 | 0.84 |
>
> **W2:** The scheduler with constant weights is mentioned but deferred to Appendix C for ablation.
>
> Thanks for this feedback. We will move this ablation study to the main paper to better clarify the optimality of the scheduler with constant weights.
>
> **W3:** Additional quantitative metrics to support t-SNE.
>
> We thank the reviewer for this valuable suggestion. To quantitatively validate our t-SNE visualizations, we evaluate the image generation quality of the DiT decoder before and after SFT. Specifically, we compute the FID scores on the MJHQ-30k dataset[1], and **the results (FID 7.82 vs. 7.84) indicate no performance degradation, which further supports our claim.** We will add this to the revised version.
>
> [1] Playground v2.5: Three Insights towards Enhancing Aesthetic Quality in Text-to-Image Generation. [https://arxiv.org/abs/2402.17245](https://arxiv.org/abs/2402.17245)
>
> **W4&Q3:** The paper focuses on Qwen-Image and MM-DiT. How transferable is your approach to other LLM-DiT architectures (e.g., PixArt-α with different text encoders, or FLUX with T5 encoders)? Have you conducted any experiments with alternative architectures to assess generalizability?
>
> Thanks for this constructive question. We select Qwen-Image as our base model because its text encoder Qwen2.5-VL is a state-of-the-art model that provides strong reasoning capacities. Moreover, it natively supports image inputs, which is necessary for reasoning-oriented editing tasks. In comparison, **the T5 encoders in FLUX or PixArt-α possess limited reasoning capabilities and lack multimodal input support.** We agree that the adaptation of alternative architectures is a promising direction and will leave it as our future work.
>
> **Q2:** How does Dual-GRPO compare to alternative approaches like (a) first optimizing the LLM with text-based rewards, then the DiT, or (b) using a single reward function for both components?
>
> Thank you for raising this interesting question. Regarding (a), this setting aligns with our staged reward scheduler, as ablated in Table 5 of our paper. Results show that the balanced co-optimization used in Dual-GRPO outperforms the staged scheduler.
>
> Regarding (b), we conduct an additional experiment using a single reward function (with a 1:1 balanced weight of semantic and aesthetic) and compare results on the T2I-ReasonBench. As shown in the table below, while a single reward improves a lot upon the baseline, component-specific reward formulation further enhances both generation accuracy and quality. We will add this to the revised version.
>
> | | Acc. $\uparrow$ | Qual. $\uparrow$ |
> | :--- | :--- | :--- |
> | Qwen-Image | 57.8 | 87.5 |
> | single reward | 68.0 | 91.9 |
> | Ours (component-specific reward) | **68.3** | **92.6** |
>
> **W5:** Discussion of computational costs and training efficiency.
>
> Thanks for this feedback. We discuss the computation costs below and will include this in the revised version.
> - **Training Cost:** We utilize 8 H200 GPUs for training. The training process consumes 2 hours for SFT activation and 36 hours for the Dual-GRPO phase.
> - **Inference Cost:** We calculate the inference latency on the WISE benchmark. Using a single A100 GPU, the CoT generation takes approximately 6 seconds, while the DiT denoising process (50 steps) requires 50 seconds.
>
> We hope these clarifications and additional results fully address your initial concerns.

---

> > ### Author Rebuttal · Reviewer_qagR · 2026-04-06
> >
> > The authors provided answered to my questions

---

> > > ### Author Response · Authors · 2026-04-08
> > >
> > > Thank you for acknowledging that our answers successfully addressed your questions. We also deeply appreciate your positive feedback and constructive suggestions, which have been very helpful in improving this work.

---

### Decision · Program_Chairs · 2026-04-30

**Decision:**

Accept (regular)

**Comment:**

All reviewers unanimously recommend acceptance for this technically solid and well-motivated paper. The authors elegantly address a fundamental limitation in current text-to-image system, treating LLMs merely as frozen text encoders, by introducing the "Think-Then-Generate" paradigm and a novel Dual-GRPO framework that successfully co-optimizes the LLM encoder and diffusion backbone. During the discussion phase, the authors provided a highly effective rebuttal with extensive new experiments, including human-judge correlation metrics, inference cost analyses, and critical ablations comparing joint versus staged optimization. These additions successfully resolved all initial reviewer concerns regarding novelty, VLM-judge circularity, and architectural design choices. Given the strong empirical improvements on reasoning-heavy benchmarks and the unanimous positive consensus, I confidently recommend this paper for acceptance.